

SciPost Phys. Lect. Notes 29 (2021)

# Machine learning and quantum devices

**Florian Marquardt**[*]

Max Planck Institute for the Science of Light and
Friedrich-Alexander-Universität Erlangen-Nürnberg, Erlangen, Germany

[*] Florian.Marquardt@mpl.mpg.de

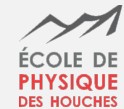

*Part of the Quantum Information Machines
Session 113 of the Les Houches School, July 2019
published in the Les Houches Lecture Notes Series*

## Abstract

These brief lecture notes cover the basics of neural networks and deep learning as well as their applications in the quantum domain, for physicists without prior knowledge. In the first part, we describe training using backpropagation, image classification, convolutional networks and autoencoders. The second part is about advanced techniques like reinforcement learning (for discovering control strategies), recurrent neural networks (for analyzing time traces), and Boltzmann machines (for learning probability distributions). In the third lecture, we discuss first recent applications to quantum physics, with an emphasis on quantum information processing machines. Finally, the fourth lecture is devoted to the promise of using quantum effects to accelerate machine learning.

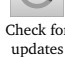

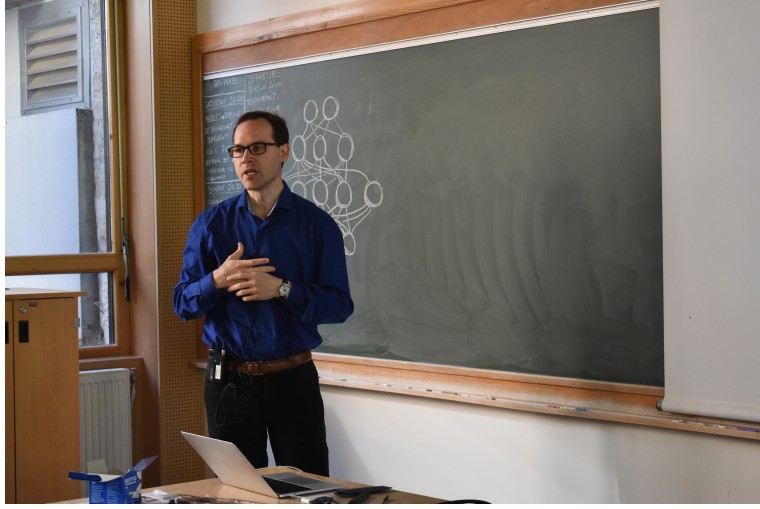

# 1 General remarks

These lecture notes cover the material of four lectures delivered in Les Houches in the summer of 2019.

The emphasis of the first two sections is on teaching the basics and some more advanced concepts of classical machine learning – sometimes illustrated in examples drawn from physics. This part relies on a lecture series that I delivered in the summers of 2017, 2019, and 2020: "Machine Learning for Physicists", at the university in Erlangen, Germany. That course runs a full semester and covers more material, although some specific examples are new to the present notes. The videos and slides for that lecture series are available via the website machine-learning-for-physicists.org. This includes links to python code for some examples. You can find more references to the classic papers of machine learning in general and deep learning in particular as a LinkMap on "Deep Learning Link Map".

The third and fourth section of these lecture notes are specifically devoted to applications of machine learning to quantum devices and to quantum machine learning, respectively – this followed the general topic of the Les Houches school: "Quantum Information Machines".

I thank the organizers of this Les Houches school as well as the enthusiastic students. In particular, however, I want to thank my graduate student Thomas Fösel, who helped me set up the first course on this topic in 2017 and whose expertise in machine learning has been of great help ever since.

# 2 A Practical Introduction to Neural Networks for Physicists

## 2.1 What are artificial neural networks good for?

During the past few years, artificial neural networks have revolutionised science and technology [1, 2]. They are being used to classify images, to describe those images in full sentences, to translate between languages, to answer questions about a text, to control robots and self driving cars, and to play complex games at a superhuman level. In science (and specifically in physics), they are being used to predict the properties of materials, to interpret astronomical pictures, to classify phases of matter, to represent quantum wave functions, and to control quantum devices. Many of these developments, especially in physics, have taken place only in the last few years, since about 2016. In the context of machine learning in physics, several good reviews [3–8] are by now available, documenting the rapidly growing field, both with respect to applications of classical machine learning methods to physics, as well as with respect to the promise of using quantum physics to accelerate machine learning.

The reasons for the recent string of successful applications are not so much conceptual developments (although they are also happening at a rapid pace), but rather the availability of large amounts of data and of unprecedented computing power (including the use of graphical processing units).

## 2.2 Neural networks as function approximators

Essentially, neural networks [2] are very powerful general-purpose function approximators that can be trained using many (i.e. at least thousands of) examples.

Let us consider a whole class of functions that has been parametrized:

$$y = F_\theta(x). \tag{1}$$

Below we will see how $F_\theta$ looks like specifically for a neural network. Suppose, in addition, we are handed some particular smooth function,

$$y = F(x). \tag{2}$$

The goal will be to approximate $F$ as well as possible by choosing suitable parameters in $F_\theta$. In the context of neural networks, we are talking about *many* parameters (hundreds or thousands), $\theta = (\theta_1, \theta_2, \ldots)$, and typically also of high-dimensional input $x$ and output $y$.

In a general sense, one can view the training of an artificial neural network as a more advanced example of curve fitting, albeit with thousands of parameters. However, it would be wrong to reduce it *only* to that description. After all, quantum many-body physics is in principle "only" about a Schrödinger equation in high-dimensional space – but in practice it brings in many new phenomena and requires new solution techniques. The same can be said about neural networks.

In many applications to empirical data, no underlying function $F(x)$ is actually known – the relation between input $x$ and output $y$ is merely specified for a large number of samples, where each sample is given by an input/output combination $(x, y)$.

In principle, the function $F_\theta$ in Eq. (1) could be constructed arbitrarily. However, we want to make sure that this representation is (i) scalable and (ii) efficient. Scalability means we need a structure that can easily be scaled up to more parameters (or higher input or output dimensions), if needed. Efficiency relates not only to the evaluation of $F_\theta(x)$, but also to the computation of derivatives with respect to the parameters $\theta$, since that is needed for training (as we will see). Neural networks fulfill both requirements, with their pairwise connections between simple units arranged in a layered structure.

## 2.3  The layout of a neural network

The basic unit of an artificial neural network is the neuron, which holds a scalar value (a real number). The operation of this neuron is simple (Fig. 1a). Its value $y$ is obtained starting from the values $y_k$ of some other neurons that feed into it, in the following manner: We first calculate a linear function of those values, $z = \sum_k w_k y_k + b$. The coefficients $w_k$ are called the "weights", and the offset $b$ is called the "bias". Afterwards, a nonlinear function $f$ is applied, to yield the neuron's value, $y = f(z)$. The points in the input space for which $z > 0$ or $z < 0$ are separated by a hyperplane $z = 0$ (Fig. 1c). This arrangement itself already constitutes an elementary neural network, a so-called (single-layer) "perceptron" that was widely investigated in the 60s for classification tasks before its limitations were fully recognized.

To obtain a nonlinear function $F_\theta(x)$ that can truly represent arbitrary functions $F(x)$, multiple layers of neurons are needed. Each neuron receives the values of all the neurons in the preceding layer, with suitable weights.

To keep the notation precise for the multi-layer case, we will now have to introduce extra indices. We denote as $y_k^{(n)}$ the value of neuron $k$ in layer $n$. Then the "weight" $w_{jk}^{(n+1)}$ tells us how neuron $k$ in layer $n$ will affect neuron $j$ in layer $n + 1$. For any neuron, we thus have the following two equations:

$$z_j^{(n+1)} = \sum_k w_{jk}^{(n+1)} y_k^{(n)} + b_j^{(n+1)}.$$

$$y_j^{(n+1)} = f(z_j^{(n+1)}).$$

The constant offset values $b_j^{(n+1)}$ are called the "biases". The output of the network is obtained by going through these equations layer by layer, starting at the input layer $n = 0$, whose

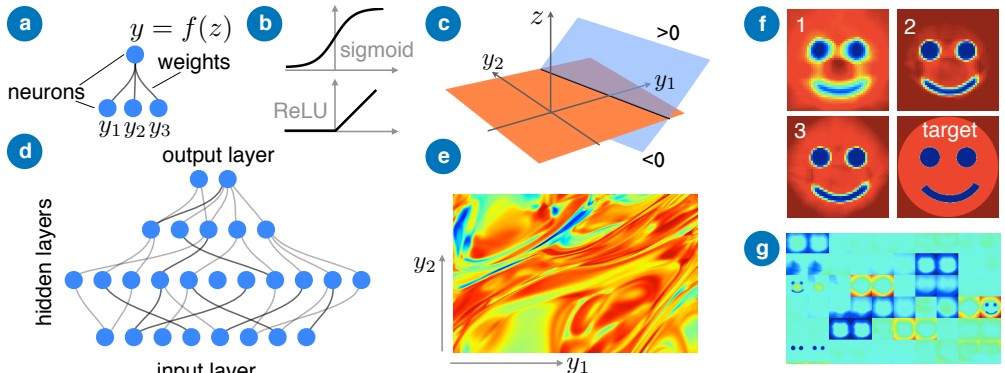

Figure 1: Structure and training of an artificial neural network. (a) Operation of a single neuron. (b) Popular nonlinear activation functions. (c) The linear weighted sum of input values, $z = \sum_k w_k y_k + b$. Applying a sigmoid to $z$ will set to 1 the output for all points in the half-space where $z > 0$, and yield 0 for all other points (with a smooth transition). (d) Structure of a neural network. (e) Output of a deep neural network with two input neurons (coordinates in the picture) and one output neuron (whose value determines the color), for randomly chosen weights. (f) Learning a scalar function of 2 variables, in this case the color values of a 2D picture, with a deep network. Panels 1,2,3 illustrate the training progress. (g) Analyzing the network's operation: each panel represents the output of a modified network where all but one of the neurons in the last hidden layer have been switched off artificially.

neuron values are provided by the user. The computational effort (and memory consumption) scales quadratically with the typical number of neurons in the layer, since there are $N^{(n+1)}N^{(n)}$ weights connecting two layers with $N^{(n+1)}$ and $N^{(n)}$ neurons. Big neural networks can quickly become memory-intensive.

It is all the weights $w$ and biases $b$ that together form the parameters of the network, and we will collectively call them $\theta$ (so $\theta$ would be a vector that contains all weights and biases). They will be updated during training.

On the other hand, the nonlinear function $f$ ("activation function") is usually kept fixed. Popular activation functions are: (i) the "sigmoid", a smoothened step-function (or inverted Fermi-Dirac distribution), $f(x) = 1/(1 + e^{-x})$; and (ii) the "ReLU", an even simpler function that is piecewise linear, $f(x) = 0$ for $x < 0$ and $f(x) = x$ for $x \geq 0$ (Fig. 1b). In recent times, the ReLU has been used predominantly, since its gradient can be calculated very efficiently and training seems to get stuck less frequently.

Neural networks are very powerful function approximators. It turns out that a single hidden layer with sufficiently many neurons can approximate an arbitrary (smooth) function of several variables to arbitrary precision [9]. Interestingly, practically any nonlinear activation function will do the job, although some may be better for training than others. However, a representation by multiple hidden layers may be more efficient, i.e. would be able to reach a better approximation with the given overall number of neurons or use fewer neurons for a given approximation accuracy (sometimes this difference can be dramatic). Such a multi-layer network is sometimes called a "**deep network**", especially if the number of layers becomes larger than a handful. Fig. 1e illustrates the complex output that can be obtained from a multilayered network whose parameters have been chosen randomly.

**Exercise: "approximating an arbitrary 1D function"** – Given a smooth function $F(x)$ of one variable, show that it can be approximated to arbitrary accuracy using only a single hidden

layer and smooth step functions (sigmoids) for the neurons in this layer. Hint: Think of a piecewise constant approximation to $F$. How do you have to choose the weights (for the input–hidden and hidden–output connections) and biases in terms of this piecewise approximation? How can you make the sigmoid steps arbitrarily sharp?

**Exercise: "the XOR function"** – The XOR function $F(x_1, x_2)$ should yield 1 for the cases $x_1 = 1, x_2 = 0$ and $x_1 = 0, x_2 = 1$ but 0 for $x_1 = x_2 = 0$ and $x_1 = x_2 = 1$ (we do not care about other input values). How can you approximate it using a network with only one hidden layer? This was an important example which could not be solved without any hidden layer (triggering a crisis in the early development of neural networks).

## 2.4 Training: cost function and stochastic gradient descent

We would like to consider some measure of the deviation between the network output and the function it is trying to approximate. To this end, we will introduce the **cost function** $C$. In the simplest case, we might just measure the quadratic deviation between the network's output $F_\theta(x)$ and the true answer $F(x)$. We first define the sample-specific cost function (depending on the specific input $x$) as

$$C_x(\theta) = |F_\theta(x) - F(x)|^2. \tag{3}$$

Subsequently, we average over all points $x$ – according to a distribution that reflects the likelihood of encountering some $x$ in real data. The averaging will automatically take place during training on the set of given training examples (see below). This yields the cost function itself:

$$C(\theta) = \langle C_x(\theta) \rangle_x. \tag{4}$$

Throughout, we have made it explicitly clear that the cost function depends on the network's parameters $\theta$.

In the scientific literature, you will often see machine learning tasks defined in terms of this high-level description, where one writes down a cost function (maybe more complicated than the one given here). For the field of machine learning, specifying a cost function is as essential as it is, for physics, to specify a Hamiltonian or a Lagrangian. It defines the problem to be solved.

Once the cost function has been defined, the basic idea is to try finding its minimum by gradient descent, in the high-dimensional space of the parameters $\theta$. It is a most remarkable fact that this simple approach to such a complex high-dimensional problem actually works very well in many cases. One of the immediate questions that spring to mind is whether one has to fear getting stuck in a local minimum. For now, let us just say that the problem exists but is not as bad as one might assume. We will come back to this issue further below, at the end of this section.

Let us now discuss how to implement the gradient descent. When you read a research paper, the steps to be explained in the following paragraphs will usually not be mentioned, because they are assumed known. Also, using modern software libraries, you may not even have to implement them yourself. However, it is very important to understand what is going on in practice, "under the hood", while training a neural network. That is because the great success of artificial neural networks depends crucially on the fact that these steps can be carried out efficiently.

In principle, gradient descent is simple. We just move along the negative gradient of the cost function (which thereby plays the same role as a potential):

$$\delta\theta_k = -\eta\frac{\partial C(\theta)}{\partial\theta_k}.$$
(5)

The parameter $\eta$ is called the **learning rate**. If it is too small, learning will proceed slowly, but if it is too large, one may overshoot the optimum. In the limit of small $\eta$, it is easy to show that the step of Eq. (5) reduces the value of the cost function: $\delta C = -\eta(\partial C/\partial\theta)^2 + O(\eta^2)$.

There are two immediate challenges connected with this approach: (i) In principle, the cost function is defined as an average over all possible inputs, which is much too expensive to calculate at each step. (ii) The cost function depends on many parameters, and we have to find a way to calculate the gradient efficiently.

The first problem is solved by averaging only over a small number of randomly selected training samples (called a "**batch**", or sometimes more precisely a "mini batch"):

$$C(\theta) \approx \frac{1}{N}\sum_{j=1}^{N}C_{x_j}(\theta) \equiv \langle C_x(\theta)\rangle_{\text{batch}}.$$

This defines the **stochastic gradient descent** method:

$$\delta\theta_k = -\eta\left\langle\frac{\partial C_x(\theta)}{\partial\theta_k}\right\rangle_{\text{batch}} = -\eta\frac{\partial C(\theta)}{\partial\theta_k} + \text{noise}.$$

The basic idea is that the noise averages out after sufficiently many small steps. That works only if $\eta$ is small enough.

## 2.5 Backpropagation

We still face the task of calculating the gradient of the cost function with respect to its parameters. Numerical differentiation (which was actually used in the early days of neural network training!) is extremely inefficient due to the large number of parameters. Fortunately, the structure of an artificial neural network allows for a much more efficient approach. The basic idea is very simple: just apply the chain rule!

For the quadratic cost function, we have:

$$\frac{\partial C_x(\theta)}{\partial\theta_k} = 2\sum_l([F_\theta(x)]_l - [F(x)]_l)\frac{\partial[F_\theta(x)]_l}{\partial\theta_k}.$$
(6)

Here $[F_\theta(x)]_l = y_l^{(N)}$ is the value of neuron $l$ in the output layer $N$. The real task is therefore to calculate the gradient of a neuron value with respect to any of the parameters:

$$\frac{\partial y_l^{(n)}}{\partial\theta_k} = f'(z_l^{(n)})\frac{\partial z_l^{(n)}}{\partial\theta_k},$$
(7)

where we have (in the case that $\theta_k$ is not among the weights or biases for this layer):

$$\frac{\partial z_l^{(n)}}{\partial\theta_k} = \sum_m w_{lm}^{(n,n-1)}\frac{\partial y_m^{(n-1)}}{\partial\theta_k}.$$
(8)

Here we see two things: First, there is obviously a recursive structure. Second, this equation can be viewed as a matrix-vector product. We now define the following matrix:

$$M_{lm}^{(n,n-1)} = w_{lm}^{(n,n-1)}f'(z_m^{(n-1)}).$$
(9)

Using the above equations, it is then easy to show that the following relation holds (if $\theta_k$ is not among the weights and biases between the layers $n$ and $n'$):

$$\frac{\partial z_l^{(n)}}{\partial \theta_k} = \left[ M^{(n,n-1)} M^{(n-1,n-2)} \cdots M^{(n'+1,n')} \frac{\partial z^{(n')}}{\partial \theta_k} \right]_l,$$

(a product of matrices applied to a vector).

This leads to the so-called **backpropagation algorithm** [10,11]: (a) Initialise the following "deviation vector" at the output layer $N$: $\Delta_j = (y_j^N - [F(x)]_j) f'(z_j^{(N)})$. (b) For each layer, starting at $n = N$, store the derivatives with respect to the weights (and biases) at that layer: $\partial C_x(\theta)/\partial \theta_k = \Delta_j \partial z_j^{(n)}/\partial \theta_k$, for all $\theta_k$ explicitly occuring in $z_j^{(n)}$. (c) Step down to the next lower layer by setting $\Delta_j^{(\text{new})} = \sum_k \Delta_k M_{kj}^{(n,n-1)}$. At the end, all of the derivatives will be known.

It is crucial that this algorithm is computationally no more demanding than the so-called forward pass (i.e. the evaluation of the network for a given input)! It also reuses the values obtained for the neuron values during the forward pass. Without the backpropagation algorithm, none of the modern applications of neural networks would have become possible. Implementing it e.g. in python takes no more than a page of code.

We can have the following useful qualitative picture for why the backpropagation alorithm works. Taking the gradients of $C$ amounts to asking for the influence that a small perturbation in one of the weights will have on the cost function. It is therefore similar to calculating a Green's function (or response function) in physics. We already know that a Green's function that measures the response of some point $f$ to a perturbation in point $i$ can be decomposed as a sum over all products of Green's functions that first connect $i$ to some intermediate point $j$ and then $j$ to $f$ – roughly $G_{fi} = \sum_j G_{fj} G_{ji}$. The same principle is used here, where we multiply the "Green's functions" of a neural network layer by layer.

## 2.6 First examples: function approximation, image labeling, state reconstruction

### 2.6.1 Approximating a function

In a first example, we try to approximate a scalar function of two variables. We choose a network with two input neurons, a set of hidden layers with more neurons each (in this case 150,150,100 neurons), and one single output neuron. We use the quadratic cost function.

To make matters more interesting, this "function of two variables" will actually be defined by an image: $F(x_1, x_2)$ will be the gray-scale value of the pixel at location $(x_1, x_2)$. After a sufficient amount of training, we can see how the original image is nicely reproduced to a good degree of approximation (Fig. 1f).

If the number of parameters were small enough, the trained network could be viewed as a compressed version of the image (in practice, this is not an efficient algorithm for image compression).

One of the important questions for neural networks is "how does it work"? Sometimes the analysis of the inner workings of a network is referred to as "opening the box". A simple approach is to artificially modify the network, e.g. by switching off neurons. In Fig. 1g, we illustrate what happens if we switch off all the neurons but one in the last hidden layer. The resulting output reveals that different neurons have learned to encode different parts of the image (e.g. only the outline of the head, or only the eyes, or sometimes everything) – in this example, we also see there is a lot of redundancy. We could probably have reduced the number of neurons and layers significantly.

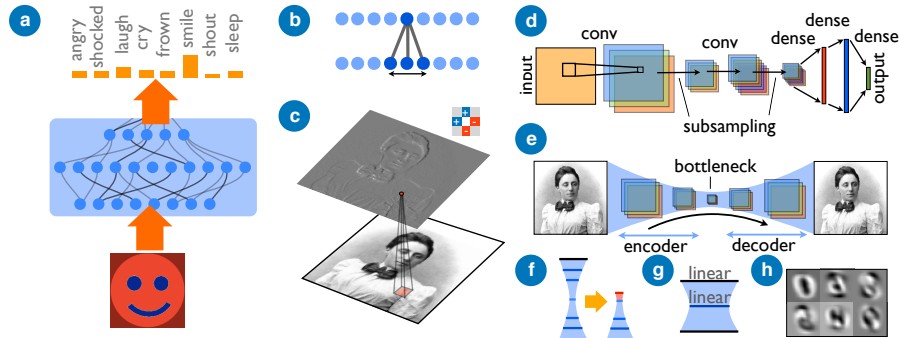

Figure 2: (a) Image classification, where the output neurons signify different labels. (b) A one-dimensional convolutional neural network, where the value of a neuron only depends on some 'nearby' neurons in a lower layer, in a translationally invariant way. (c) For 2D images, application of such filters can extract features like contours (the filter is shown as an inset). (d) A full-fledged CNN, with convolutional steps, several channels (indicated as multiple images overlaid over each other), subsampling, and finally a transition to densely connected layers. (e) An autoencoder tries to reconstruct the input after having compressed the information into a few latent variables inside a bottleneck layer. (f) After training, the 'encoder' part of an autoencoder can be repurposed for a specific classification task. (g) A fully linear autoencode will find a projection onto the most important principal components of the set of input vectors. (h) The six most important principal components of the MNIST handwritten digits images.

### 2.6.2 Image classification

One of the main applications of neural networks is image classification. Given an image, the network is asked to label it (e.g. as a "giraffe" or a "dolphin" etc.). In 2012, a deep neural network was able to beat all other algorithms in the so-called "ImageNet" competition [12], and since then such networks have surpassed even humans in their accuracy to properly recognize and label images.

The input layer contains as many neurons as there are pixels in the image, with the neurons set to the pixels' brightness values. In the output layer of an image classification network, each neuron is responsible for a different category (label). It is supposed to represent the likelihood that the input image falls into that category (Fig. 2a). To obtain a suitably normalized distribution of output values, one uses the so-called "**softmax**" activation function. Suppose we have already calculated the values $z_j$ for the output layer from the linear superpositions of the previous layer's values. Then the new value $y_j$ of output neuron $j$ is defined in the following manner (which depends on all the other neurons, in contrast to what we encountered before):

$$y_j = f_j(z_1, \ldots, z_M) = \frac{e^{z_j}}{\sum_{k=1}^{M} e^{z_k}}. \tag{10}$$

This is automatically normalized and non-negative. It can be viewed as a multi-dimensional generalization of the sigmoid.

The "correct" output distribution has a value of 1 in a single spot, for the neuron that corresponds to the correct label for the given image. All other values are zero. This is also known as a "**one-hot encoding**".

How should we define the cost function? In principle, we could take the quadratic deviation between the one-hot encoding and the network output. However, we are essentially comparing

two probability distributions (the "true", one-hot distribution, and the network output). For that purpose, there exists an alternative, the so-called **categorical cross-entropy**

$$C = -\sum_j P_j^{\text{target}} \ln P_j. \tag{11}$$

You can show that minimizing this with respect to $P_j$ will yield $P_j = P_j^{\text{target}}$. In our case, $P_j^{\text{target}}$ is the one-hot correct label (= 1 for exactly one of the $j$), and $P_j = y_j^{(N)}$ are the output neuron values. This cost function is preferable, since its gradients are less likely to become small.

A well-known test-case for image labeling is the MNIST dataset, where more than 50000 images ($28 \times 28$ pixel) of handwritten digits have been labeled according to the digit they represent. More complex datasets are also available freely for download, such as the ImageNet dataset.

Training on the MNIST set with a modest neural network (with $28^2$ input neurons, 30 hidden neurons, and 10 output neurons) already can yield a nice performance, with about 3% of error. However, one has to take care: in a naive approach, the accuracy on the training data is getting ever better while one is repeatedly going through the same set of training samples. At the same time, the accuracy on test data (that the network has never seen) actually decreases again after some point. The reason is an important and well-known problem: so-called "**overfitting**". The network essentially memorizes the training examples so well that it becomes very good on them, paying attention to the slightest details. However, for examples it has never seen, this reduces its performance, because it can no longer properly generalize. There are several solutions to this. First, one may artificially augment the training data, generating new samples e.g. by rotating and scaling images (since the labels typically should not change under these operations). Another solution is to keep a small set of samples separate as so-called "**validation data**", and to constantly monitor the network's performance on these samples during training. Once the performance starts to decrease again, one has to stop ("**early stopping**"). Another, very powerful solution is to introduce noise. The noise prevents the network from overfitting if it is sufficiently strong. In practice, this means randomly switching off a small fraction of neurons during training (or multiplying them with random Gaussian values). That strategy, which was invented only rather recently, is called "**drop-out**".

Importantly, if you generate your data always fresh (i.e. the network never sees a sample twice), then there is no danger of overfitting. This will be feasible e.g. for random training samples generated through simulations or experiments, provided that each simulation or experimental run is not too costly.

### 2.6.3 A first quantum physics example: State reconstruction

Let us try to come up with a relatively simple but useful example in quantum physics. We could, for example, teach a neural network to time-evolve a quantum state, where the input would be a quantum state at time zero (e.g. for a discrete basis: $\Psi_j(t = 0)$) and the output during training would be set to the time-evolved state $\Psi_j(t)$ at some later time $t$. If training is carried out on many randomly chosen states $\Psi$, this would teach the network effectively to implement the unitary time-evolution operator $e^{-i\hat{H}t}$. However, since that operator is linear, the network itself would not need any nonlinearity and the example is therefore maybe a bit too trivial. A slightly more interesting variant would be to provide not the state but various expectation values of observables and to ask for their time evolution. However, since these are linear in the density matrix and that also evolves linearly, it is still not a challenging problem.

Consider another problem, that of quantum state reconstruction. Given several identical copies of a quantum state $|\Psi\rangle$, and a set of projective measurements on those copies, try to figure out the state from the measurement results.

Let us turn this into a simple challenge for a neural network, where it will be the network's task to provide us with an estimate of the quantum state based on the measurement outcomes. To make things concrete, we imagine measuring copies of a qubit state in several basis directions (denoted by projectors $\hat{P}_1, \hat{P}_2, \ldots, \hat{P}_M$). These directions have been fixed (by us) beforehand, e.g. we might measure a few times along the z-axis and a few times along the x-axis etc. For any given quantum state and any given "experimental run", that procedure yields a string of $M$ random measurement results $x_1, x_2, \ldots$, where $x_j = 1$ with probability $p_j = \langle \Psi | \hat{P}_j | \Psi \rangle$ and $x_j = 0$ otherwise. These will be the input to the network. We will then ask the network to provide us with an "estimate of the state". This is best done by asking it to output the density matrix, which for a qubit can be represented as a three-dimensional real-valued Bloch vector, $\vec{y} = \langle \Psi | \hat{\vec{\sigma}} | \Psi \rangle$.

So far, so good. Now we have to make an important choice: how do we set up the cost function, i.e. how do we punish the network if it deviates from the real state? After all, even the best network will not be able to guess the state perfectly, since it has to rely only on a limited set of binary measurement outcomes (whereas the space of all states is continuous). It makes sense to demand the network's output to be as close as possible to the real Bloch vector, since that ensures the predictions for all the three qubit observables are as correct as possible. Let us here choose the simple quadratic deviation. You should be aware, however, that the network might sometimes output unphysical states, i.e. Bloch vectors of magnitude larger than 1. If we want to avoid this at all costs, we should correspondingly modify the cost function, to yield very large (infinite) values outside the physical range. In this simple example, we will not go that far.

How do we choose the quantum states during training? This choice is very important, since it will determine the network's responses. If we only ever were to show states during training that are either pointing up or down in the z-direction, the network will learn this and also assume any other state will be of this kind. Let us therefore choose states with Bloch vectors uniformly distributed on the Bloch sphere.

In fact, the network's task is related to Bayes reasoning. Given the a-priori probability of having certain states (here defined by the distribution of training samples!), and given the observed measurement outcomes, what is the distribution of likely states after the extra information produced by the measurement has been taken into account? Since we are, however, only asking for a single state (Bloch vector), the network should try to pick the state that minimizes the cost function under the new probability distribution that has been obtained using the Bayes rule. This distribution over states $\Psi$, given the measurement outcomes $x$, is:

$$P_{\text{new}}(\Psi) = \frac{P(x|\Psi)P_{\text{prior}}(\Psi)}{\int P(x|\Psi')P_{\text{prior}}(\Psi')d\Psi'}. \tag{12}$$

The denominator involves averaging over all states $\Psi'$ according to the prior distribution. Evaluating this and then finding the optimal choice for the predicted Bloch vector is not a trivial task, and the network (in order to be optimal) would have to discover all of this just from the training samples.

## 2.7 Making your life easy: modern libraries for neural networks

Even just a few years ago, one might have implemented the neural network code oneself. While this is fine for the basics, it can get cumbersome when trying to implement the latest advances. Nowadays, the situation has changed dramatically. There are all kinds of libraries that simplify the implementation considerably. These include libraries like tensorflow, PyTorch, and others. Here we will illustrate the power of these approaches using **keras**. This is a widely used high-level python library that interfaces with lower-level libraries like tensorflow (and is actually

automatically included in any installation of tensorflow nowadays; you can import commands from tensorflow.keras).

This is all it takes in **keras** to produce a network with two hidden layers (layer sizes, from input to output, are 2,30,20,1):

```
net=Sequential()
net.add(Dense(30,input_shape=(2,),
    activation='relu'))
net.add(Dense(20,activation='relu'))
net.add(Dense(1,activation='linear'))
```

Nothing more is needed! The first line creates a new network called 'net', to which layers are then added one by one. "Sequential" refers to the standard layered network layout we have been discussing without exception, though keras also can be used to produce more advanced designs, where the network forks into branches or there are connections between layers further apart. "Dense" means densely connected layers, in contrast e.g. to convolutional layers that we will discuss below. We have set the activation functions to ReLU for the two hidden layers, but linear (no activation function) for the output layer. Any combination is possible, including also 'sigmoid'. The softmax function mentioned above would be indicated as 'softmax'.

We should still specify the cost function (called "loss" in **keras**). This is done in a step labeled 'compilation'. It is here that the gradients are produced using symbolic differentiation, to be used during training. Moreover, in this step the network's weights are initialized randomly. Let us just set up the simplest kind of cost function, the usual quadratic deviation between training samples and network output:

```
net.compile(loss='mean_squared_error',
    optimizer='adam')
```

One alternative loss would have been 'categorical_crossentropy' (see above). In the second line, we have also selected a so-called 'optimizer'. This refers to the method used during gradient descent. Instead of the simple stochastic gradient descent introduced above, we have chosen a more modern advanced technique called 'adam'. This is an adaptive scheme, where essentially the learning rate for each parameter is chosen automatically to provide for faster convergence. It is one of the most popular choices right now, though there may be occasional cases where it is less stable than the standard stochastic gradient descent.

Training data will be provided in the form of an array of inputs $x$ (of dimension batchsize $\times M_{in}$, where $M_{in}$ is the number of input neurons) and outputs $y$ (of dimension batchsize $\times M_{out}$). A single line runs through the whole batch and updates the network's parameters:

```
net.train_on_batch(x,y)
```

This returns the current value of the cost function. Repeated application (preferably to randomly selected fresh data) will be needed to train the network. Finally, to evaluate the network on any given data $x$, we would write

```
y=net.predict_on_batch(x)
```

Now $y[j,:]$ will contain the output vector obtained for the j-th sample in the batch, $x[j,:]$. If you think of 2D data (like pixels in an image), use numpy's `flatten` and `reshape` commands to convert into (and back from) 1D vectors.

## 2.8 Exploiting translational invariance: convolutional neural networks

Often, the meaning of an image does not depend on translations – e.g. when a handwritten letter is shifted a bit. In other words, we are facing a problem with translational invariance,

similar to what is often the case in physics. In a physics scenario of this kind, the response $A(x)$ at point $x$ to a perturbation at $x'$ only depends on the displacement from that point. Mathematically, this is represented by a convolution: $A(x) = \int G(x - x')F(x')dx'$.

**Convolutional neural networks** (CNNs) [13,14] exploit translational invariance by restricting significantly the structure of the neural network weights. The weights now only depend on the distance. Let us first consider the 1D situation (Fig. 2b):

$$w_{ij}^{(n+1,n)} = w^{(n+1,n)}(i-j). \tag{13}$$

The function $w^{(n+1,n)}(i-j)$ would be called the **'kernel'** or the **'filter'**. It is cut off beyond a certain distance, i.e. set to zero for $|i-j| > d$. Most importantly, you should note a tremendous reduction in the amount of weights that have to be stored and updated during training. We switched from $M^2$ weights (if both layers have $M$ neurons) to a fixed number $2d + 1$ that does not even depend on the size of the layer! The standard network layout we have discussed above is referred to as '**densely connected**' layers, in contrast to CNNs, which have a sparse weight matrix.

In 2D, each neuron sits at a specific pixel in an image, so we could label it by two discrete coordinates, $i = (i_x, i_y)$. Then the kernel depends on $i_x - j_x$ and $i_y - j_y$, i.e. it is given as a small 2D array of dimensions $(2d + 1) \times (2d + 1)$.

The linear part of such a network's operation corresponds exactly to what happens in a photo-editing program when applying linear filters: $z_i^{(n+1)} = \sum_j w^{(n+1,n)}(i-j)y_j^{(n)} + b^{(n+1)}$. These can be used to smoothen an image or to highlight contours (Fig. 2c). The difference here though is not only the subsequent application of a nonlinear activation function, but more importantly the fact that the CNN filters will be learned automatically during training.

One of the amazing insights that CNNs have provided is the connection to the visual cortex in the brain. It turns out that in our brain the lowest layers, right after the retina, effectively implement filters that detect edges and the orientation of those edges. The same functionality arises in deep CNNs during training, practically irrespective of the task for which they have been trained (task-specific details emerge in higher layers).

It is deep CNNs that underlie the success of neural network at image classification tasks. In such applications, two additional features are implemented: **channels** and **subsampling**. Often, an image already has several color channels. In addition, in higher layers, it becomes useful to store different features in separate channels (Fig. 2d). This requires introduction of an extra channel index $c$, such that each neuron is now labeled in the form $(i, c)$. Then, we would have:

$$z_{(i,c)}^{(n+1)} = \sum_j w_{cc'}^{(n+1,n)}(i-j)y_{(j,c')}^{(n)} + b_c^{(n+1)}. \tag{14}$$

If you think about this, it is a hybrid between the operation of densely connected layers (with respect to the channel indices) and single-channel CNNs.

Often, it is useful to reduce the resolution when passing towards higher layers. For example, one may just subdivide an image into $3 \times 3$ patches and replace each of those with a single pixel whose value is the average (similar to block decimation in the real-space renormalization group). This is called subsampling. Finally, to carry out the actual classification of an image, at some late stage one may switch from CNN back to densely connected layers. This can simply be done by taking all the neurons in all channels and arranging them back into one big vector (a so-called "flattening" operation).

In a framework such as keras, setting up a full-fledged CNN requires only a few lines (try `Conv2D` instead of `Dense`, `AveragePooling2D` for subsampling, and `Flatten` for the transition to dense layers).

## 2.9   Unsupervised learning: autoencoders

Up to now, we have dealt with what is called "**supervised learning**": when providing the training samples, both the input and the desired correct output have to be specified.

However, suppose you just have a large amount of data and you do not yet know whether there are any specific patterns to be discovered in this data. One example may be a large set of unlabeled images. Can a network discover on its own that these images represent different categories (e.g. "cats", "dogs", and "birds" ?).

It turns out that there is a surprisingly simple way to force a network to develop an understanding of the most important features of a set of unlabeled training data. The basic idea is to require the network's output to (approximately) reproduce its input: $y = F_\theta(x) \approx x$. This seems a trivial task, until you learn about an extra requirement. One of the layers in the middle of the network has very few neurons, much less than the number of input and output neurons. This layer is sometimes referred to as a "bottleneck", through which the information has to pass. In order to succeed at this task, the network has to compress or encode the relevant features of the input, pass it through the **bottleneck**, and then reconstruct the input based on that limited amount of information. This can only work well if the whole set of all inputs is highly structured (it could never work, e.g., if the inputs are completely random vectors).

Such a network is called an "**autoencoder**" (Fig. 2e) [15, 16]. The layers below the bottleneck form an "**encoder**", while the layers afterwards form a "**decoder**". The neurons in the bottleneck layer are called "**latent variables**". They represent the main features that the network has learned in this **unsupervised** (or, more precisely, "self-supervised") fashion. This is an example of the broader field of **"representation learning"** [17].

Note that the autoencoder structure can be used for different tasks than unsupervised feature extraction. For example, it can be trained to denoise images [18]: One can feed as input an image that has had noise added to it artificially, while the output is still the clean image. The network will learn (as well as possible) to get rid of the noise, and this will work eventually for noisy images it has never seen before during training (provided the noise is similar in structure to what it has encountered before). The same works for partially occluded images. Another nice example is colorization of images: For training, a large number of color images are obtained, but the input is always taken to be the gray-scale version of the image. The network will learn to automatically fill in the colors. In this way, black-and-white movies can be turned into realistically colored movies (although there is no guarantee that the colors are indeed correct, because sometimes there are simply several equally plausible options).

Once an autoencoder has been trained, it can easily be used as the basis for solving another task, e.g. classification. The idea is to re-use the already trained encoder layers, and then add one or more layers on top of it. During subsequent training, only these additional layers need be trained, and they will much more quickly converge to a good solution, since the basic features have already been extracted in the encoding stage (Fig. 2f).

**Example: Linear autoencoder (principal component analysis)** – We now turn briefly to the simplest possible autoencoder: A single hidden bottleneck layer and no activation functions – i.e. a purely linear network! What are the weights that it will find? As you will see, this is a highly instructive example [15].

To keep the following discussion simple, let us also assume the input vectors have zero mean, $\langle x \rangle = 0$, in which case we will not need biases in the network. Then the value of neuron $j$ in the hidden layer is

$$y_j^{(1)} = \sum_k w_{jk}^{(1,0)} x_k. \tag{15}$$

This can be interpreted as a projection of the input vectors onto some other set of vectors $\left| v_j \right\rangle$,

modulo normalization, of the kind $w^{(1,0)}_{jk} = \langle v_j | k \rangle$. In the output layer, we have $y^{(2)}_l = \sum_j w^{(2,1)}_{lj} y^{(1)}_j$. This should be approximately equal to $x$. In summary, we want to minimize:

$$C = \left\langle \left| x - w^{(2,1)} w^{(1,0)} x \right|^2 \right\rangle = \left\langle |x - \tilde{w} w x|^2 \right\rangle. \tag{16}$$

Here the matrix $\tilde{w} \equiv w^{(2,1)}$ is of size $M_\text{out} \times M_\text{hidden}$, and $w \equiv w^{(1,0)}$ is of size $M_\text{hidden} \times M_\text{out}$. In other words, $A = \tilde{w} w$ must be "as close as possible to the identity", even though it is a matrix of rank at most $M_\text{hidden}$.

Minimizing $C$ with respect to $A$ is a well-defined problem. The essential quantity in carrying out the average will be the correlation matrix of the input vectors:

$$\rho_{lj} = \left\langle x_l x_j \right\rangle, \tag{17}$$

(in a physics analogy, think of the density matrix, $\rho_{lj} = \left\langle \Psi_l \Psi^*_j \right\rangle_\Psi$).

Using this definition, we can rewrite Eq. (16) as

$$C = \text{tr}\left( \rho - 2A\rho + A^t A \rho \right). \tag{18}$$

We now choose to write the trace in the basis of eigenvectors of the symmetric matrix $\rho$:

$$C = \sum_j \rho_{jj} - 2A_{jj}\rho_{jj} + \sum_{j,k} A^2_{kj} \rho_{jj}. \tag{19}$$

To minimize this over an arbitrary $A$, we should choose $A_{kj} = 0$ for all $k \neq j$, and make $\sum_j A_{jj}(2 - A_{jj})\rho_{jj}$ as large as possible. Each term in this sum would have its maximum at $A_{jj} = 1$ (since $\rho_{jj} > 0$). However, since $A$ has at most rank $M_\text{hidden}$, we can only choose that many diagonal elements $A_{jj}$ to be nonzero (and equal to 1). We obviously have to choose those for which the eigenvalues $\rho_{jj}$ are maximum.

In other words, this linear autoencoder has to implement the projector onto the subspace that contains the eigenvectors with the largest eigenvalues of the correlation matrix $\rho$ of the inputs. Calling these (orthonormal) eigenvectors $|v_j\rangle$, we have for the output of this network:

$$|y\rangle = \sum_{j=1}^{M_\text{hidden}} |v_j\rangle \langle v_j | x\rangle, \tag{20}$$

where the eigenvectors have been ordered, with the largest eigenvalues first.

In data science, the decomposition of the correlation matrix of inputs into its eigenvectors is known as **"principal component analysis"**. Strictly speaking, the latent variable neurons of this linear autoencoder need not correspond to the projections onto individual eigenvectors (the overall operation of the network only needs to implement the subspace projector). However, if desired, this can be "fixed" by demanding that there are no correlations between latent variables, $\left\langle y^{(1)}_j y^{(1)}_k \right\rangle = 0$ for $j \neq k$. Such a constraint can be added to the cost function, for example in the form $C_\text{new} = C_\text{old} + \sum_{j \neq k} \left\langle y^{(1)}_j y^{(1)}_k \right\rangle^2_\text{batch}$. This kind of requirement can also be useful in the context of arbitrary (nonlinear) autoencoders.

**Exercise: Denoising autoencoder**   Train a neural network to get rid of noise in images that show a randomly placed circle of random size. Use convolutional layers with downsampling for the encoder, and convolutional layers with upsampling for the decoder (use `UpSampling2D`). Generate random training images and feed a noisy version of each image into the autoencoder as input, while defining the original image as the target. Vary the challenge by producing other sorts of training images, with more complicated shapes! Instead of simple noise, try to obscure

the original image by deleting pieces (e.g. setting all pixels to zero in randomly chosen small squares).

## 2.10 Some warnings for the enthusiastic beginner

After witnessing the impressive success of artificial neural networks, it is tempting to become a bit too enthusiastic. You should realize that there are several challenges:

- Training a neural network is a highly nonlinear and stochastic process (not well-understood theoretically). Training several times from scratch on the same training data (starting from random weights) will usually result in networks with different weights, even though their performance may be similar.

- Results depend strongly on the quantity and quality of training data.

- Applying a neural network (or more generally machine learning techniques) to data is no substitute for basic understanding.

- Interpretation of the results requires care. A neural network is like a black box, and extra effort is needed to understand its inner workings.

# 3 Advanced Concepts: Reinforcement Learning, Networks with Memory, Boltzmann Machines

## 3.1 Discovering strategies: reinforcement learning

### 3.1.1 Introduction

So far, we have dealt with a simple scenario: a neural network is shown many training examples, where the correct answer (e.g. the correct label for an image) is already known. That scenario is known as "supervised learning".

In a sense, this describes a knowledgeable teacher training a student, but in a simplistic way. The student essentially learns to imitate the teacher's answers. In the best case, the student may be able to extrapolate from these examples in a modest way, but it will likely never surpass its teacher in any substantial aspect. The power of the approach comes from the student being infinitely diligent and patient, but not from any creativity.

In contrast, let us consider what we expect from a really talented student or from a scientist. We would hope that they are able to discover good novel solutions to problems on their own, without having been provided with answers to a large range of rather similar training problems. In real life, this creative approach to problem-solving requires the following. First, there is a lot of trial and error. Second, once we stumble on a good solution, we have to be able to recognize it as such. That means there should be a criterion to decide whether one solution is better than another. Third, if we have discovered a set of good solutions, we may want to recombine them in novel ways, to find even better solutions.

In the field of machine learning, this approach is known as "**reinforcement learning**" (RL, for short) [19, 20]. It represents the most promising approach to future general artificial intelligence [21], especially when combined with deep neural networks. Recent years have brought spectacular applications of deep reinforcement learning: a neural network can learn to play video games purely by observing the screen and the score [22] or it can learn to play sophisticated board games like Go [23, 24], becoming better than the best humans.

The general RL setting is a control problem. Imagine a robot that interacts with the world around it. In RL language, the robot is an "**agent**", and the world around it is the



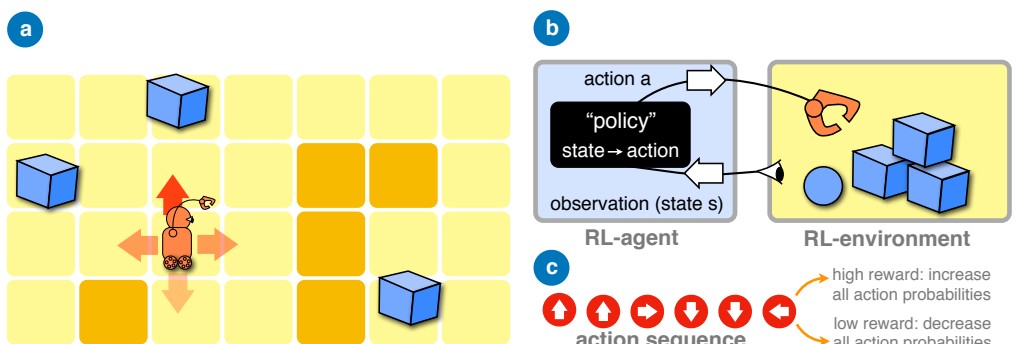

Figure 3: Reinforcement learning. (a) A robot roaming around a grid world and trying to pick up boxes is one simple example of a reinforcement learning problem. (b) The general scheme: In each time step, the observed state $s_t$ of the environment is used to choose the next action $a_t$, according to the agent's policy, represented by the probability $\pi_\theta(a_t|s_t)$. (c) Basic principle of policy gradient. Given a certain action sequence, the action probabilities for all the actions involved in this particular sequence will be increased if the reward turns out to be high.

"**environment**". The robot can manipulate objects in the world and move around. It can also observe its environment and choose its subsequent actions depending on the observations – this is an example of feedback control. The situation is displayed in Fig. 3.

The mapping from the observed state of the environment to the next action is called "**policy**". The policy effectively defines the strategy that the robot implements.

### 3.1.2 Policy gradient approach

To make things concrete, we will now describe one of the oldest RL approaches, the so-called "**policy gradient**" method [25] – even nowadays this is one of the most powerful techniques, with suitable variations and extensions. Here, the policy is probabilistic. Let us imagine time $t$ is discrete, and in each time step an observation is taken and a next action is chosen. If the observed **state** of the environment at time $t$ is $s_t$, then the policy is a probability distribution over all possible **actions** $a_t$ given that state:

$$\pi_\theta(a_t|s_t).$$

The actual next action will be chosen randomly according to this distribution. We typically have in mind a discrete set of actions $a_t$. For example, if the robot can move around on a grid, $a_t = N, S, W, E$ might indicate motion by one "step" into the corresponding direction.

The subscript $\theta$ for the policy indicates that the policy depends on a set of parameters $\theta$. These will be updated during training. In the advanced cases we are interested in, the policy will be represented by a neural network, and $\theta$ will be its parameters (weights and biases).

We still have to define the goal of this game. This is done via "**rewards**". At each time step, a reward $r_t$ is provided, depending on the state $s_t$ and the action $a_t$ that was taken. For the robot, we might want it to pick up boxes, and therefore assign a reward $r_t = +1$ for each time it picks up a box. The sum of all rewards in a given time interval then will be equal to the total number of boxes that have been picked up. This sum of rewards is called the "**return**" $R$ (used in the sense of "return on investment"):

$$R = \sum_{t=1}^{T} r_t. \tag{21}$$

Since the policy is probabilistic, and also the environment may have stochastic dynamics, the return $R$ will fluctuate from run to run, even if the policy is kept fixed. We are interested in the expectation value of the return, averaged over all runs (or "trajectories"):

$$E[R] = \sum_\tau p_\theta(\tau) R(\tau). \tag{22}$$

Here we have introduced $\tau$ as a label for a **trajectory**: $\tau = (s_1, a_1, s_2, a_2, \ldots, s_T, a_T)$. The probability $p_\theta(\tau)$ of observing this trajectory depends on the policy. We will now assume the environment can be modeled as a Markov process, where the next state $s_{t+1}$ only depends on the current state and the current action, and there is a transition probability $P(s_{t+1}|a_t, s_t)$. Then, the trajectory's probability can be factorized in the following manner:

$$p_\theta(\tau) = \Pi_{t=1}^T P(s_{t+1}|a_t, s_t) \pi_\theta(a_t|s_t). \tag{23}$$

This is a string of conditional probabilities, alternating between the action choices of the agent and the transitions of the environment (for the purposes of this expression we may set $P(s_{T+1}|a_T, s_T) = 1$). Note that the assumption of a Markovian environment is much less restrictive than it may sound at first: We can have arbitrarily complicated dynamics if the total number of degrees of freedom in the state space is large enough. The policy only depends on the *observable* degrees of freedom (i.e. $\pi_\theta(a_t|s_t) = \pi_\theta(a_t|s_t')$ if the states $s_t$ and $s_t'$ coincide in the observed quantities). These may be only a small subset and their dynamics can be non-Markovian, since it is driven by the unobserved parts of the environment – the usual reason for having non-Markovian dynamics in nature.

The basic idea of the policy gradient approach is to optimize the expected return via gradient ascent with respect to the policy parameters $\theta$:

$$\delta\theta = +\eta \frac{\partial}{\partial\theta} E[R]. \tag{24}$$

This is symbolic notation: More precisely, $\theta$ is a whole vector containing all parameters, and this equation should be read as $\delta\theta_j = +\eta\frac{\partial}{\partial\theta_j}E[R]$, for all $j$. One of the most important features of Eq. (23) is that the dependence on the policy $\theta$ does not enter the environment's transition probabilities $P$. This will enable us to take the gradient with respect to $\theta$ without actually having any explicit knowledge of $P$ – it would be very hard for most real environments to construct a detailed model for their dynamics. Since the RL approach is independent of having such a model, it is called "**model-free**". That sets it apart from other numerical methods for optimizing control, such as GRAPE (used for quantum control, with an explicit model for the Hamiltonian). This is the reason we cannot easily use a deterministic policy, because there the effect of any change in the policy at time $t$ would affect the subsequent environment dynamics, and to understand the consequences for the return, we would have to differentiate through the unknown environment dynamics.

We now evaluate the gradient. First, we note that the gradient of $p_\theta(\tau)$ can be written in the following way:

$$\frac{\partial}{\partial\theta} p_\theta(\tau) = \sum_t \frac{\partial_\theta \pi_\theta(a_t|s_t)}{\pi_\theta(a_t|s_t)} \Pi_{t'} P(s_{t'+1}|a_{t'}, s_{t'}) \pi_\theta(a_{t'}|s_{t'}). \tag{25}$$

This comes about because taking the gradient of a product means differentiating each factor separately and then adding up the results (take a moment to understand it). We have already re-arranged terms such that it becomes obvious this can be further simplified to

$$\frac{\partial}{\partial\theta} p_\theta(\tau) = p_\theta(\tau) \sum_t \frac{\partial}{\partial\theta} \ln \pi_\theta(a_t|s_t). \tag{26}$$

Overall, after inserting back into Eq. (22), we obtain a surprisingly simple expression:

$$\delta\theta = \eta \frac{\partial}{\partial\theta} E[R] = \eta E[R(\tau) \sum_t \frac{\partial}{\partial\theta} \ln \pi_\theta(a_t|s_t)]. \tag{27}$$

This is the main result for the policy gradient approach. What it means in practice is the following. We run through a trajectory and note all the actions we took. In the end, we calculate the return. We change the policy parameters according to the logarithmic gradient of the policy, evaluated for these actions, multiplied by the return. *All* the actions that have been taken are made more likely, but more so if the return is larger (the norm is conserved, you can try to show this yourself). Averaged over many trajectories, this has the effect of reinforcing the "good" actions, i.e. actions that have been taken primarily in high-return trajectories.

### 3.1.3   Extremely simple RL example: training a random walker

Let us look at what must be the simplest possible RL example, a biased random walker, whose probability $\pi_\theta(a_t = +1)$ to go "up" can be varied during training (Fig. 4a). Note that this policy does not depend on any observed state, so there is no feedback yet in this example. The goal of the walker is to reach as far as possible from the origin, i.e. make $R = x(T)$ as large as possible. Of course we know that the optimal strategy is simply to always go up. However, it is very instructive to see how this strategy is reached.

Let us use a sigmoid for $\pi_\theta(+1) = (1 + e^{-\theta})^{-1}$. During training, we need the policy gradients, to evaluate the central equation (27). Do the math to show that they are:

$$\partial_\theta \ln \pi_\theta(+1) = 1 - \pi_\theta(+1), \tag{28}$$

$$\partial_\theta \ln \pi_\theta(-1) = -\pi_\theta(+1). \tag{29}$$

As a consequence, we find:

$$\sum_t \partial_\theta \ln \pi_\theta(a_t) = N_+ - T\pi_\theta(+1). \tag{30}$$

Here $N_+$ is a fluctuating quantity, namely the number of "up" steps: $N_+ = \sum_{t=1}^{T} \delta_{a_t,+1}$. Its expectation value is $\bar{N}_+ = T\pi_\theta(+1)$. In other words, Eq. (30) measures by how much this number, for a particular trajectory, exceeds the average. To get the update $\delta\theta$, according to Eq. (27), we only have to multiply by the return $R(T)$ and take the expectation value. This will yield a positive update for $\theta$ if trajectories with more "up" steps than average yield an enhanced return. For our scenario, this should be the case. Let's see whether the math bears out this expectation!

For the return, we obtain $R = x(T) = \sum_t a_t = N_+ - N_- = 2N_+ - 1$. Now we see that this example is so simple that the update equation can be obtained analytically (a very rare case):

$$\delta\theta = \eta E[R \sum_t \partial_\theta \ln \pi_\theta(a_t)] = \eta E[(2N_+ - 1)(N_+ - \bar{N}_+)]. \tag{31}$$

Rewriting slightly and using $E[N_+ - \bar{N}_+] = 0$, we find that the update just depends on the variance of $N_+$:

$$\delta\theta = \eta E[2(N_+ - \bar{N}_+)^2] = 2\eta T\pi_\theta(+1)(1 - \pi_\theta(+1)). \tag{32}$$

In the last step we used the formula for the variance of a binomial distribution.

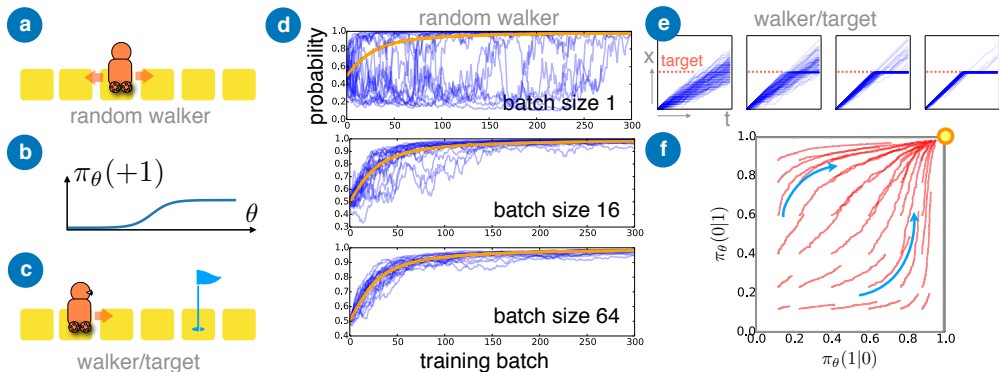

Figure 4: Simple illustrations of reinforcement learning. (a) A random walker moving either left or right in each time step, where the reward will be determined according to the distance covered to the right. (b) Dependence of the probability $\pi_\theta(+1)$ for moving right on the policy parameter $\theta$. (c) The walker/target example, where the walker has to learn to stop when on target. (d) Learning progress for the random walker, using policy gradient: we display the probability to move right (which should ideally converge to 1). Increasing batch sizes lead to smoother learning behaviour. (e) Walker/target example, with a sample of trajectories $x(t)$, displayed for increasing learning progress (from left to right). Eventually, the walker learns to stop on target (rightmost panel). In this plot, the target was always placed at the same location $x$, though it has random locations during training. (f) Training evolution of the probabilities to move when not on target, $\pi_\theta(1|0)$, and to stop when on target, $\pi_\theta(0|1)$. The fixed point is indicated, representing the optimal policy.

Does this update equation make sense? First, we note that it is always positive. And an increase in $\theta$ also increases the probability $\pi_\theta(+1)$ to go up. This is exactly what is needed to increase the return!

Second, we find that the update vanishes in the extreme cases. When the walker *always* goes up already, no further increase of the probability is necessary or possible, so this is fine. On the other hand, when the walker *always* goes down, nothing happens either (which is bad). The walker is stuck with the worst possible strategy. The reason for this is that then there is not even a single trajectory that deviates from the expected behaviour, and thus the walker never even gets a chance to see larger returns. In RL jargon, this is called a lack of "exploration". Whenever that is a problem, a typical solution is to introduce random actions once in a while (i.e. not follow the policy all the time).

The largest update steps are obtained at $\pi_\theta(+1) = 1/2$, i.e. when the walker is unbiased. Then the fluctuations of $N_+$ are largest, and the walker efficiently explores all possibilities.

The resulting training progress is shown in Fig. 4d.

**Exercise: Training a walker** – Implement the stochastic training update numerically, by drawing the random $N_+$ according to a binomial distribution, and using the update equation in the form (31) – but *without* taking the expectation value $E[\dots]$ (just evaluate for a particular trajectory, i.e. a particular value $N_+$). Plot the evolution of $\pi_\theta(+1)$ during training, and repeat several times to observe the stochastic nature of training. Show numerically that for a sufficiently small learning rate $\eta$, we obtain the behaviour expected from the averaged equation (plot the curve expected from this average equation for comparison)! Empirically, for which values of $\pi_\theta(+1)$ are the fluctuations in the update the largest?

### 3.1.4 Simple RL example: Walker reaching a target

We now change the scenario to include feedback: a walker that wants to find a target site and stay there (Fig. 4b). The observed state is either 0 (most of the time) or 1 (on the target site). The goal of the game is now to have the walker spend as much time as possible on the target. For simplicity, we slightly revise the walker's actions: it can now either stay ($a_t = 0$) or move up ($a_t = +1$). We will assume the target is somewhere at a positive position $x^*$, so that it can be reached at all. This position will be chosen at random before the start of each trajectory. Again, it is clear to any human neural network after a few seconds of thinking what is the best strategy: Move up as fast as possible in the beginning, but stop once the target site is reached. In terms of the policy, this means: $\pi_\theta(a_t = 1|s_t = 0) = 1$, $\pi_\theta(0|1) = 1$, and zero for the two other policy probabilities. Let us see how this policy is reached!

One can probably once more obtain an analytical solution (I have not tried it). However, it is also fun to implement this example numerically. We still do not really need a neural network: there are only two independent policy probabilities (due to normalization), so it is enough to introduce sigmoids with parameters $\theta_0$ for $\pi_\theta(1|0) = (1 + e^{-\theta_0})^{-1}$ and $\theta_1$ for $\pi_\theta(0|1) = (1 + e^{-\theta_1})^{-1}$.

Implementing this example numerically means: (i) simulate a trajectory stochastically; (ii) for this trajectory, evaluate the quantity $\sum_t \partial_{\theta_0} \ln \pi_\theta(a_t|s_t)$ [and likewise for $\theta_1$], where the derivative has been calculated analytically beforehand; (iii) record the return $R$ (the number of timesteps during which the walker was on target); (iv) apply the policy gradient update rule to both $\theta_0$ and $\theta_1$.

The progress during training is shown in Fig. 4d. The walker becomes ever better at moving quickly at first and then staying on target. These figures illustrate that the procedure works as expected. During training, one observes an upward drift of both the probability to "stay on target" and to "move when not on target". This flow reaches the ideal policy that we have identified before (Fig. 4e).

### 3.1.5 Quantum physics RL example

We now want to apply RL techniques to a realistic scenario from quantum physics. The purpose is to illustrate that we can already obtain valuable results based on nothing more but the policy gradient approach introduced above. In order to make best use of the capabilities of RL, we will naturally choose a situation involving feedback. We consider a quantum system that is controlled by a neural network based on the results of some measurements that have been performed on that system. That description covers a rather wide class of problems, many of them extremely interesting for applications in modern quantum technologies.

For the purposes of this example, we will keep the quantum system itself simple. It is a single mode of a cavity, i.e. a harmonic oscillator. However, this mode decays, and it can be measured, so we will need to describe its quantum dissipative dynamics. In addition, the cavity mode can be acted upon, e.g. via an external drive. The situation is displayed in Fig. 5a. This example is more valuable than one might think at first sight, given the important role that cavities play in scenarios like qubit coupling and readout as well as a potential quantum memory and even as a qubit.

In RL, it is standard to think of a discrete time, which we can implement here by choosing a small time step $\Delta t$ (this may be subdivided further into even smaller time steps for the physics simulation, if needed).

The actions that the neural network can take in this situation are in principle described by continuous values. It might want to adjust the drive amplitude $\alpha_{\text{in}}$ of a beam entering the cavity, or it might be allowed to control something else, like the frequency $\omega_L$ of the drive beam, or even the strength of some additional nonlinear term in the cavity Hamiltonian. All

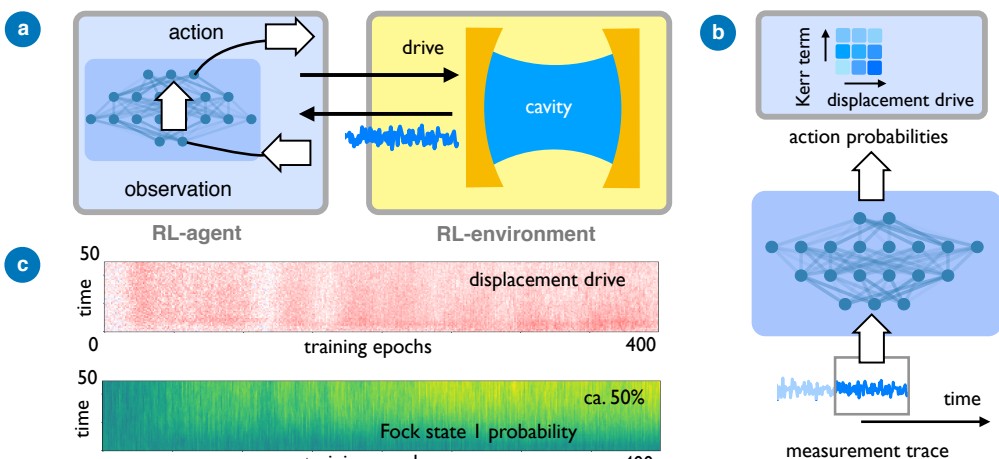

Figure 5: Reinforcement learning in quantum physics. (a) The quantum feedback setting in our example, a cavity that is observed and driven. (b) The network converts a measurement trace into probabilities for all the available actions. In this picture, continuous control amplitudes are assumed to be discretized to yield discrete action choices. For two control parameters, this results in an array of possible parameter combinations, each of which represents one action. (c) The training progress, illustrated via the drive amplitude (red means higher amplitudes), and via the resulting probability for Fock state 1 in the cavity. Obviously the network becomes better at stabilizing this Fock state as the training progresses.

of these are easily accessible to the RL approach. In fact, the network does not need to be changed for any of these choices, it is only the interpretation of the actions that changes. The appropriate RL variant to apply here would be continuous RL (where the network outputs continuous values which are interpreted as the center of some Gaussian distribution). However, to keep things simple and in line with the preceding discussion, we will merely discretize the continuous values. For example, if there is only a drive amplitude to care about, we will predefine a number of discrete amplitudes and label them by an integer: $\alpha_{\text{in}} = \alpha_a$, $a = 1 \ldots N_\alpha$. These are then the action choices whose probabilities $\pi_\theta(a|s)$ are output by the network. If we have more than one continuous control parameter, we would have to let $a$ label a discrete set involving all possible combinations of values, which quickly becomes cumbersome (and would be a good reason to switch to continuous RL), see Fig. 5b.

The input to the network (i.e. the observed state $s$) will be the measurement trace. In principle, the most sophisticated approach at this point would be to use a network with memory (recurrent network, see below), and to feed in one measurement result per time step. However, to keep things simple, we will not use a recurrent network. Instead, we will always present most recent values of the measurement signal as input to the network ($T_{\text{msmt}}$ data points, which thus defines the number of input neurons). In this way, the network can react at least to a finite time interval of the fluctuating signal. That may allow it to average the signal if needed or perform some more sophisticated interpretation of the time trace.

Finally, we have to choose the reward function. Again, there are many possible choices, all of which can be selected without any change in the underlying algorithm. Here, we will aim for quantum state stabilization, i.e. the reward is the overlap of the cavity's state $\hat{\rho}_t$ at time $t$ and a fixed given state: $r_t = \langle \Psi | \hat{\rho}_t | \Psi \rangle$. Another interesting choice would be the overlap with a particular subspace of states. The return $R = \sum_{t=1}^{T} r_t$ will favor a policy that goes to the target state rather quickly.

In principle, all of this could be applied in an experiment. The prerequisites would be

sufficiently fast control hardware, where a neural network is able to access quickly the measurement results and produce the feedback signal. However, for the purpose of these lecture notes, which had to be prepared without the benefit of a laser or a microwave generator, we will simulate the dynamics on a computer.

The Hamiltonian of a cavity mode driven at resonance is most suitably described in a frame rotating at the cavity frequency. In this frame, it only contains the drive: $\hat{H} = i\sqrt{\kappa}(\alpha_{\text{in}}\hat{a}^{\dagger} - \alpha_{\text{in}}^{\dagger}\hat{a})$, which would lead to a Heisenberg equation of motion $\dot{\hat{a}} = \sqrt{\kappa}\alpha_{\text{in}}$. (Here $|\alpha_{\text{in}}|^2$ would be the number of drive photons per unit time impinging on the cavity) The unitary dynamics of the cavity's quantum state $\hat{\rho}$ is then determined by $i\dot{\hat{\rho}}_{\text{unitary}} = [\hat{H}, \hat{\rho}]$. Moreover, the decay of photons at the cavity decay rate $\kappa$ is described by a Lindblad term, $\dot{\hat{\rho}}_{\text{decay}} = \kappa\mathcal{D}[\hat{a}]\hat{\rho}$, where we adopt the usual definition $\mathcal{D}[\hat{R}]\hat{\rho} = \hat{R}\hat{\rho}\hat{R}^{\dagger} - \frac{1}{2}(\hat{R}^{\dagger}\hat{R}\hat{\rho} + \hat{R}^{\dagger}\hat{R}\hat{\rho})$. Finally, we have to treat the stochastic measurement signal. We can do this using the quantum jump trajectories approach. Given a measurement operator $\hat{A}$, we find a noisy classical measurement trace:

$$X(t) = \sqrt{\kappa'}\langle\hat{A} + \hat{A}^{\dagger}\rangle + \xi(t), \tag{33}$$

where $\langle\hat{A} + \hat{A}^{\dagger}\rangle = \text{tr}[\hat{\rho}(\hat{A} + \hat{A}^{\dagger})]$ and $\xi(t)$ is a stationary Gaussian white noise stochastic process, $\langle\xi(t)\xi(0)\rangle = \delta(t)$. The induced stochastic dynamics of the state $\hat{\rho}$ is:

$$\frac{d}{dt}\hat{\rho}_{\text{msmt}} = \kappa'\mathcal{D}[\hat{A}]\hat{\rho} + \sqrt{\kappa'}(\hat{A}\hat{\rho} + \hat{\rho}\hat{A}^{\dagger} - \langle\hat{A} + \hat{A}^{\dagger}\rangle\hat{\rho})\xi(t). \tag{34}$$

The measurement operator has to be selected according to the physical situation. Suppose we do a homodyne measurement of the linear amplitude of the field leaking out of the cavity. For clarity, assume the left mirror is described by $\kappa$, while we measure the field leaking out of the right mirror at a rate set by $\kappa'$. Then we would choose $\hat{A} = \hat{a}$. On the other hand, if we had available a more sophisticated setup where a QND measurement of the photon number inside the cavity can be performed, we would have $\hat{A} = \hat{a}^{\dagger}\hat{a}$. That could be achieved by a Kerr coupling between cavity modes [26].

Using these equations, we can implement a physics simulation of our driven, dissipative cavity. This simulation will evolve the system's state forward by an amount $\Delta t$, based on the current value of the drive amplitude. After this short step, the neural network is queried again. Given the measurement trace, which has been updated according to Eq. (33), the network will decide on the next action probabilities. One of these actions is selected, and the next physics simulation step will be executed. This procedure is performed until the fixed end $T$ of the trajectory, before the network's parameters are updated according to the policy gradient approach. For efficiency, all of this is done in a parallelized fashion, on a batch of trajectories that are processed simultaneously (so there is always a set $\hat{\rho}_j(t)$ of states to keep track of, where $j = 1 \ldots N_{\text{batch}}$).

Fig. 5 shows some results obtained using this approach. As our goal, we have chosen to stabilize the Fock state with one photon in the cavity, $|\Psi\rangle = |1\rangle$. To facilitate this, we assume a weak QND measurement of the photon number inside the cavity. The control simply consists in a linear drive (as explained above). One clearly observes the improvement of the policy during training, as the Fock state probability increases. The observed values of the Fock state probability in this example are already beyond what could be obtained simply from a coherent state, even if its displacement were chosen optimally. Many extensions of this example are possible, by choosing different goals (i.e. rewards), control knobs (e.g. controllable Kerr terms inside the Hamiltonian), readout approaches. In any case, it is not quite trivial to analyze the performance of the RL approach e.g. with respect to analytically constructed feedback control strategies.

### 3.1.6 Q learning

We now briefly describe an alternative RL approach, different conceptually from the policy gradient method. All the other present-day RL techniques can essentially be traced back to either one of those two techniques or are hybrids between the two concepts.

The idea of Q learning [27] is to introduce a so-called **quality function** $Q(s_t, a_t)$ that is the expected future return if one takes action $a_t$ in state $s_t$:

$$Q(s_t, a_t) = E[R_t]. \tag{35}$$

Here $R_t = \sum_{t'=t}^{T} r_{t'} \gamma^{t-t'}$ is the so-called **discounted** future return, with a discounting factor $\gamma < 1$. This means immediate rewards are considered more important ($\gamma = 0$ would result in a greedy strategy that always tries to maximize the next reward, without concern for the long-term consequences). Eq. (35) is nontrivial: The expectation on the right-hand-side is taken over all trajectories (beginning at the present time $t$) that follow the current policy. The policy in Q learning depends on Q itself. It simply consists in always choosing the action $a$ that maximizes $Q(s, a)$. In this sense, Eq. (35) is a recursive definition of (or implicit equation for) Q. We can make this more obvious by rewriting it in the form of "**Bellmann's equation**":

$$Q(s_t, a_t) = r_t + \gamma E[R_{t+1}] = r_t + \gamma \max_a Q(s_{t+1}, a), \tag{36}$$

where $s_{t+1}$ is the state reached from $s_t$ by executing action $a_t$. This is obtained by inserting the definition of $R_t$, collecting all terms $t' > t$, and noting that *their* expectation value is exactly the Q function evaluated at the optimal action $a$ for the new state $s_{t+1}$ (up to an extra factor $\gamma$).

This is still not tractable. However, we can iteratively improve an estimate for the Q function by using the following **Q learning update rule**, which ensures we come ever closer to a solution of Eq. (36):

$$Q^{\text{new}}(s_t, a_t) = Q^{\text{old}}(s_t, a_t) + \alpha(\text{RHS}_{Q=Q^{\text{old}}} - \text{LHS}_{Q=Q^{\text{old}}}). \tag{37}$$

Here RHS and LHS refer to the right-hand and left-hand side of Eq. (36), and $\alpha \ll 1$ is a small positive number, which determines the speed of the iterative improvement.

What happens in practice is the following: At first, the Q function becomes large directly at states $s$ that give a large immediate reward. In subsequent steps of the update rule, this also affects nearby states $s'$, since one can reach $s$ from any of those. In this way, large values of the Q function tend to spread through state space, in a diffusion-like process.

In advanced applications, $Q(s, a)$ is represented by a neural network, and the update rule is implemented by training the network to approximate the new value. Q learning has been used successfully in many cases. One recent impressive example was training a network to play Atari video games, where the state $s$ consisted in a combination of the last few video frames from the game and the actions $a$ were the simple discrete controls of the form "move left" etc.

We finally mention a related concept, the so-called **value function**, that measures the expected future return only depending on a state $s$. The defining equation looks superficially the same as before, but we now assume that the next action will be chosen according to the current policy, instead of being prescribed:

$$V(s_t) = E[R_t]. \tag{38}$$

This concept, of a value function, has been combined with the policy gradient approach to yield so-called "actor-critic" RL methods. The basic idea there is to always compare the true overall return with the expected return given the current state – obviously, a sequence of action choices

that yielded only a moderate return despite starting from a high-value state cannot have been very good.

The field of deep reinforcement learning is developing very rapidly, with many extensions and hybrid variants of different algorithms. A relatively recent review is provided in [20].

## 3.2 Mimicking observed probability distributions: Restricted Boltzmann Machines

The Boltzmann machine [16,28] is an example of a machine learning tool that is very directly linked to the statistical physics of spin systems. It is also important because it can be generalized to the quantum case.

The basic task of the Boltzmann machine is to mimick an observed probability distribution $P_0(v)$ of values $v$. In the applications of interest, the values $v$ are high-dimensional (e.g. images or measurement results obtained from many measurements on a quantum many-body system). The key words here are "mimick" and "observed". We are not given access to the functional form of $P_0(v)$. Rather, we can only observe many samples $v$ drawn from this distribution. On the other hand, we also do not want to produce an approximation to this high-dimensional function $P_0(v)$ either. Rather, we want to be able to *sample* from the same distribution efficiently.

The basic idea is to set up a statistical model whose Boltzmann distribution can be adapted to approximate $P_0(v)$. The model energy $E_\theta$ depends on parameters $\theta$, which will be changed during training of the Boltzmann machine.

One crucial ingredient of Boltzmann machines is the existence of "hidden" variables $h$. The full configuration of the model at any time is described by specifying both $v$ and $h$ together, and the Boltzmann distribution we are talking about is a joint distribution:

$$P(v,h) = \frac{e^{-E_\theta(v,h)}}{Z}, \tag{39}$$

where $Z = \sum_{v,h} e^{-E_\theta(v,h)}$ is the partition sum, needed for normalization. Obviously we have set $k_B T = 1$, which just amounts to a rescaling of the energy. In a physical implementation of a Boltzmann machine, $E$ would be a dimensionless energy, rescaled by the thermal energy. In the end, we want to tune $\theta$ such that

$$P(v) = \sum_h P(v,h) \approx P_0(v). \tag{40}$$

For brevity, we have denoted as $P(v)$ the distribution over visible unit configurations $v$, and correspondingly we will write $P(h)$ for the marginal distribution $P(h) = \sum_v P(v,h)$. A more precise notation would be $P_v(v)$ and $P_h(h)$, but it will always be clear from the argument which distribution we refer to.

This arrangement, with hidden variables, makes it possible to evaluate the statistics of the model efficiently, as we will see below – provided one chooses a particular architecture, the so-called **restricted Boltzmann machine**. This has an energy given by

$$E(v,h) = -\sum_i a_i v_i - \sum_j b_j h_j - \sum_{i,j} v_i w_{ij} h_j. \tag{41}$$

Note the absence of $v-v$ or $h-h$ coupling terms, which makes this a *restricted* model. In the typical approach, the values $v_i$ and $h_j$ are binary (0 or 1). In other words, we are dealing with an Ising model of a particular restricted form, but with arbitary $v-h$ couplings. It is these couplings $w$ (as well as the 'magnetic fields' $a$ and $b$) that form the parameters $\theta$ of the model and which have to be trained.

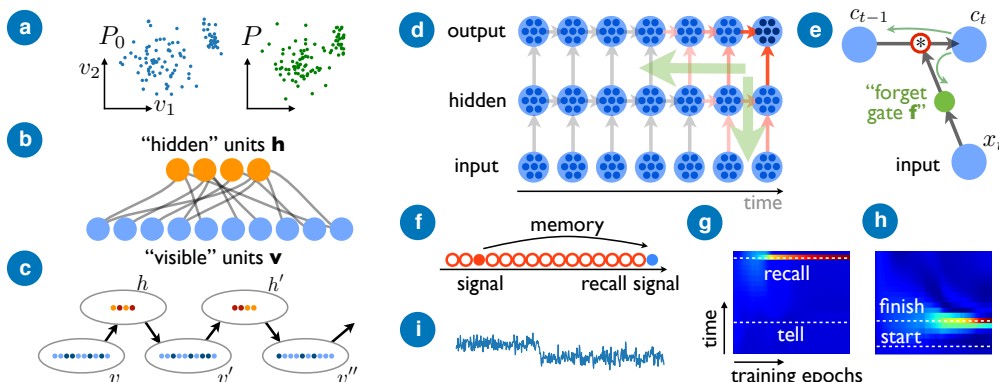

Figure 6: (a) The goal of a Boltzmann machine is to learn to sample from an approximation $P$ to an observed probability distribution $P_0(v)$. (b) A restricted Boltzmann machine, with connections between visible and hidden units. (c) During training, starting from an observed sample $v$, a Monte Carlo Markov chain is produced according to the current statistics of the Boltzmann machine. (d) Structure of a recurrent neural network, with feedforward connections in time. (e) A "forget gate" neuron as part of a long short-term memory (LSTM) network. (f) A typical challenge requiring long memory times. (g) A signal triggers the recall of a number presented earlier to the network. The training progress is shown here. (h) A network learns to count down from an arbitrary number (input at "start"). (i) A typical application of recurrent networks in physics: analyzing fluctuating measurement time traces.

In the end, we want to minimize the deviation between the target distribution $P_0(v)$ and the Boltzmann machine thermal distribution. Let us measure this deviation by the categorical cross entropy,

$$C = -\sum_v P_0(v) \ln P(v) \tag{42}$$

introduced above in the context of image recognition. Since $v$ is high-dimensional, there is no hope of actually carrying out the sum (if $v$ consists of $N$ units, the sum has $2^N$ terms). However, we can formally take the derivative with respect to the parameters. A lengthy calculation yields a comparatively simple result. For example, the derivative of the energy $E_\theta(v,h)$ with respect to the weight $w_{ij}$ generates the combination $v_i h_j$. As a result, we find:

$$-\frac{\partial}{\partial w_{ij}} C = \sum_v P_0(v) \frac{\partial}{\partial w_{ij}} \ln P(v) = \left\langle v_i h_j \right\rangle_{P_0} - \left\langle v_i h_j \right\rangle_P . \tag{43}$$

The two terms on the right-hand-side are defined as:

$$\left\langle v_i h_j \right\rangle_{P_0} \equiv \sum_{v,h} v_i h_j P(h|v) P_0(v) \tag{44}$$

and

$$\left\langle v_i h_j \right\rangle_P \equiv \sum_{v,h} v_i h_j P(h|v) P(v). \tag{45}$$

Here we have introduced the conditional probability,

$$P(h|v) = \frac{P(v,h)}{P(v)} . \tag{46}$$

To evaluate these expressions, we need a way to sample from the distribution $P(v)$, as well as to sample $h$ given $v$ according to the conditional probability $P(h|v)$. This is the challenge we address below. On the other hand, sampling over the observed empirical distribution $P_0(v)$ is easy, because we are being provided samples $v$ accordingly (that was the starting point of the whole task).

In general, sampling from a given distribution $P(s)$ (for any model with some configurations $s$) can be performed using a Monte Carlo algorithm. Any Monte Carlo algorithm is constructed as a stochastic Markov process, where the transitions between states $s$ have been chosen to fulfill detailed balance: $P(s \to s')/P(s' \to s) = P(s')/P(s)$ for all pairs of states $s, s'$. In particular, if the target is to obtain a Boltzmann distribution, we will have $P(s')/P(s) = \exp(E(s) - E(s'))$, where again we have used an energy rescaled by $k_B T$, like above.

In the present situation, we will slightly modify the standard Monte Carlo approach, by exploiting the special structure of our problem: the distinction between visible units and hidden units. Consider a Markov chain that starts from some visible unit configuration $v$, then jumps to some hidden unit configuration $h$, goes back to some other $v'$, etc. It keeps alternating between visible and hidden configurations. We define the transition probabilities as the conditional probabilities $P(h|v)$ and $P(v|h)$ that can be obtained from the underlying Boltzmann distribution $P(v, h)$. Then it is easy to check that that detailed balance holds:

$$\frac{P(h|v)}{P(v|h)} = \frac{P(h)}{P(v)}. \tag{47}$$

As a consequence, this Markov chain converges to a steady-state distribution that, for both visible units and hidden units, is equal to the respective marginal distribution $P(v)$ and $P(h)$. As an aside we note that the full Boltzmann distribution $P(v, h)$ is realized as the distribution of pairs $(v, h)$ composed of a visible configuration and the hidden configuration that it reaches in one Monte Carlo update [since $P(v, h) = P(h|v)P(v)$].

To actually implement the Monte Carlo step, we need to calculate the conditional probabilities. A brief calculation reveals

$$P(h|v) = \frac{e^{-E(v,h)}}{ZP(v)} = \Pi_j \frac{e^{z_j h_j}}{1 + e^{z_j}}, \tag{48}$$

with

$$z_j = b_j + \sum_i v_i w_{ij}. \tag{49}$$

The most important fact about Eq. (48) is that it is a *product* of probabilities, one for each hidden unit. In other words, we can sample the new values $h_j$ independently, and the probability for $h_j = 1$ is simply $\sigma(z_j)$, where $\sigma$ is the sigmoid activation function. Monte Carlo sampling of a Boltzmann machine thus consists in two steps: calculate probabilities the same way you would calculate the new neuron values for a densely connected pair of layers (with sigmoid activation), and then sample binary values $h_j = 0/1$ according to those probabilities. The step back, from $h$ to $v$, proceeds analogously (with a $z_i' = a_i + \sum_j w_{ij} h_j$).

Finally, let us return to the task of evaluating the weight update for the Boltzmann machine, Eq. (43). There are two terms, one involves sampling from the target distribution $P_0(v)$, the other requires sampling from $P(v)$. The first task is easy, by definition, since we are provided with samples from $P_0$. The second task seems hard, since we have to run a lot of Monte Carlo steps to converge to the steady state distribution. However, there is a trick we can use. If the Boltzmann machine is already close to the target distribution, $P_0(v) \approx P(v)$, then a sample $v$ from $P_0$ will be almost as good as a sample from $P$. We can then get even closer to the $P$ distribution by doing a few Monte Carlo steps starting from this sample. In practice, the

simplest approach is to take a single extra pair of steps: $v \to h \to v' \to h'$. Then $v'$, obtained in this way, can serve as a good approximation to having a sample from $P$. In this way, the right-hand side of the update equation can be approximated as:

$$\left\langle v_i h_j \right\rangle_{P_0} - \left\langle v'_i h'_j \right\rangle_{P_0}, \tag{50}$$

where the second term, written out explicitly, is:

$$\left\langle v'_i h'_j \right\rangle_{P_0} = \sum_{v,h,v',h'} v'_i h'_j P(h'|v') P(v'|h) P(h|v) P_0(v). \tag{51}$$

This approach is called "contrastive divergence". As emphasized above, the approximations involved become better when $P$ finally approaches $P_0$.

In this way, training a Boltzmann machine has been reduced to sampling from the target distribution $P_0$ and executing a few Monte Carlo steps for any given sample $v$.

The Boltzmann machine is a particular solution for the general task of *learning to sample* from an observed distribution (which is only defined via the training samples, and is not given explicitly). Besides the Boltzmann machine, there are other, more recent approaches that solve this task. **Variational autoencoders** [29] are a version of autoencoders which enforce the distribution of latent variables to be particularly simple and fixed (e.g. a normal multi-dimensional Gaussian), such that one can simply sample from this latent distribution and then use the decoder to produce valid samples that are distributed according to the observed training distribution. **Generative adversarial** networks [30] solve a similar problem by having a generator network that tries to produce samples which a detector network is no longer able to distinguish from real training samples (although here it is not guaranteed that the generator reproduces correctly the full distribution of samples).

## 3.3 Analyzing time traces: recurrent neural networks

The study of dynamics defines much of physics. Observing the dynamics of a system results in time traces, often with fluctuations (e.g. due to measurement imprecision). To analyze them with a neural network, one may hand the whole time series (with $T$ time steps) as input to the network. However, that usually implies fixing the time interval $T$ in advance, because it is connected to the network structure. One simple way around this would be to use convolutional neural networks, where the translational invariance (now with respect to time) is exploited. But this comes with a catch: the size of the filters (kernels) in such a network will determine the time-scale over which the network's memory works.

The alternative are so-called "**recurrent neural networks**", i.e. networks that have built-in memory. Basically, at each time the network not only receives fresh external input (as was the case in all the settings we discussed so far), but it also receives internal input, from the values that the neurons had at the previous time step. External and internal inputs are processed together to calculate the new, updated neuron values. In this way, in principle, recurrent networks can keep memory over arbitrarily long time spans. Training proceeds by presenting both an input time-series and the corresponding correct output time-series. Importantly, the weights are not themselves time-dependent, so the number of training parameters does not grow with the time interval $T$ that is considered. In this way a given trained recurrent network can be applied to arbitrarily long time series (in the same way that a convolutional network can be applied to arbitrarily sized images).

When training such a network, taking the gradient of the cost function will not only step down layer by layer (as in usual backpropagation) but also back in time (Fig. 6d). This can involve a lot of steps back in time, only limited by the total time interval. It was realized already in the 90s that backpropagation involving many layers or time steps can result in

problematic behaviour. In each step, the deviation vector is multiplied by a matrix, so e.g. $\Delta_{t-1} = M^{(t-1,t)}\Delta_t$ for backpropagation in time. Since that matrix can have eigenvalues larger or smaller than unity, this can lead to exponential growth or vanishing of the gradient vector $\Delta_t$. What this means is that the influence of a weight change at some early time on the network's response at some late time is either vanishingly small or exponentially large. This leads to problems in learning, especially for situations which require memory to be preserved over long times.

It was recognized by Hochreiter and Schmidhuber in 1997 [31] that this problem can be circumvented. They pointed out that a typical application scenario often looks like this (Fig. 6f): a memory is created but then remains irrelevant for a long time (during which time it need not be accessed). Only much later a certain external signal triggers recall of the memory. If that is the case, it is a smart idea to not touch the memory most of the time, i.e. to make read-out or write-in depend on external stimuli. As we will show below, this then avoids the exponential growth or vanishing of gradients.

To implement this in practice, so-called "**gating neurons**" are introduced. Their purpose is to calculate, based on the current external input, whether the memory needs to be accessed or not. Let us discuss this first for the simplest case, that of a "forget gate" neuron, which determines whether the memory should be erased. Assume a neuron carries a value $c_{t-1}$ at time step $t-1$, and we want to decide whether to keep that value. We can do this by writing $c_t = f \cdot c_{t-1}$, where $f \in [0,1]$ is the value of the gating neuron. That value, in turn, has been calculated from the external input $x_t$ (or from some lower layer), e.g. as $f_t = \sigma(wx_t + b)$. In the simplest case, $c, f, w, b$ would be scalars, but in practice we will be talking about whole layers of neurons. Then we would introduce suitable indices, to have $c_{j,t}$, $f_{j,t}$, $w_{jk}$, $b_j$, and $x_{k,t}$. Note that, for the first time, we are multiplying neuron values!

When backpropagation is applied in such a case, the product rule splits the gradient into two branches, one of which steps down to the lower layer (through the forget gate neurons), and the other goes back further in time (Fig. 6e):

$$\frac{\partial c_t}{\partial \theta} = \frac{\partial f}{\partial \theta} c_{t-1} + f \frac{\partial c_{t-1}}{\partial \theta}. \tag{52}$$

Gated read and write operations are implemented in a similar way. In that context, one distinguishes the memory content of a neuron (the $c_t$ above) and its output value (that is fed into higher layers or as output to the user). Furthermore, one can make the forget/read/write gate neuron's values also depend on the output values of neurons in the same layer, taken from the earlier time step $t-1$ (instead of just the lower layer inputs, denoted $x_t$ in the example above).

All of this results in a structure that is called "**long short-term memory**" (**LSTM**) [31]. The label "short-term memory" is to set this apart from the true long-term memory that would be encoded in the network's weights $w$ that have been modified during training. Short-term memory, by contrast, is the memory retained for the duration of a specific task (a single run, with $T$ time steps).

We do not list the detailed formulas required for implementing the various LSTM gates here, since frameworks like tensorflow and keras automatically will provide you with LSTM implementations ready to use. Just use an LSTM layer instead of `Dense`. This layer keeps track of its internal memory state, passing that state forward in time. The input to such a network now has to be of dimension batchsize × timesteps × $M_{in}$. The output is either of dimension batchsize × $M_{out}$, yielding only the output at the final time step (if the option `return_sequences` is set to `False`), or batchsize × timesteps × $M_{out}$ (if `True`), yielding the full sequence.

One of the main applications of recurrent neural networks has been for language translation: every sentence can be seen as a time-series of words. A recurrent encoder network is first used

to go through the original sentence, step by step, building up an internal representation of the meaning of the full sentence. Afterwards, a recurrent decoder network is used to produce the translated sentence word for word: In each step, it has as input available the information from the encoder as well as the sequence of words it has produced so far (or a suitable internal representation it has built from that).

It is worthwhile to mention a rather recent new approach that has had great success in this domain: "Transformer networks" [32] get rid of the recurrent nature, i.e. they do not process sentences word by word. Rather, the sentence is represented like a database, with each word an entry in the database (containing information about the word itself but also about its position in the sentence). The transformer network then generates database queries (matching keys and returning values) in order to transform the database, paying attention simultaneously to various parts of the sentence. This is currently the state-of-the-art in machine translation and general natural language processing.

**Exercise: Training an LSTM to add numbers**    Train an LSTM to perform addition, turning a sequence of the type "23+18=???" into "23+18= 41". Hint: convert each digit (as well as the special characters '+','=','?') into a one-hot-encoded binary string, for example "3" yields "0001000000000", and this becomes the input to the network at that particular time step.

# 4 Applications of Neural Networks and Machine Learning for Quantum Devices

In this section we will review a few characteristic examples of how machine learning might be applied to improve the performance of quantum devices. Some of these examples have already been realized in first proof-of-principle experiments, while others represent theoretical studies pointing the way to future experiments. We do not pretend the following to be a comprehensive review. Rather, it is our goal to give a representative sample of current research in this field.

## 4.1 Interpreting measurement outcomes

The readout of quantum states in modern quantum devices is a fertile challenge for neural networks. For example, in weak continuous measurements, given a noisy measurement trace, the network can help to extract the maximum amount of information possible. In projective measurements of quantum many-body systems, a network can help to represent the underlying quantum state.

Machine learning approaches are particularly helpful in the presence of non-idealities like extraneous noise and nonlinearities. Using suitable training samples, a network can learn to overcome these challenges. In that way, it can become better than the default approach to any given measurement problem, which relies on idealized assumptions.

**Weak qubit measurements** – For the case of weak measurements, a nice experimental example has been realized recently. It illustrates for the first time the application of neural networks to the weak measurement of a driven superconducting qubit. The standard dispersive readout of a qubit works by coupling it to a microwave cavity. Sending a microwave beam through the cavity, one can detect the phase shift that depends on the qubit state. In the experiment of the Berkeley group [33], a network was trained to analyze the resulting weak measurement trace (voltage vs. time). After preparing the qubit, it is continuously driven and simultaneously weakly monitored. Finally, a strong projective measurement is applied. The task of the network is to predict the probability for obtaining a certain projective (strong)

measurement outcome $y_t \in \{0, 1\}$ at a time $t$, $P(y_t|y_0, a, b, V_0, V_1, \ldots, V_t)$, given the prior observed fluctuating trace $V_0, V_1, \ldots, V_t$ of the weak continuous measurement, and given the projective measurement basis $b$ (as well as the initial state determined by $y_0 \in \{0, 1\}$ and a subsequent qubit preparation pulse $a$). A recurrent neural network (LSTM) is able to properly learn the dissipative quantum dynamics of a continuously measured qubit from a million experimental measurement traces. It is particularly noteworthy that the network has no notion of quantum mechanics to begin with, i.e. it learns all of the dynamics purely by example.

**Interpreting error syndromes** – In quantum error correction, an essential step is to employ collective qubit measurements in order to check whether an error has occured – without projecting the state of the logical qubit. This is known as error syndrome detection. The most important category of quantum error correction approaches are stabilizer codes. Among those, the surface code represents an easily scalable variant, where the probability of having an irreversible error decreases exponentially in the size of the qubit array. However, given a syndrome (i.e. a pattern of unexpected measurement outcomes in the surface code array), the challenge is to deduce the most likely underlying error and, consequently, the correct way to undo this error. The syndrome is essentially a 2D image, as is the underlying error (the locations of the qubits that have been flipped by the noise). Thus, this is a task well suited for neural networks, and this insight has been exploited in a series of works (see [34–36] for early examples).

**Extracting entanglement** – The logarithmic negativity probably represents the practically most useful quantity for describing the entanglement between two subsystems A and B. However, measuring it experimentally is a challenge, since it does not correspond to any simple observable. Rather, arbitrarily high moments of the (partially transposed) density matrix are needed, which translates experimentally into a large number of copies of the system that have to be measured after applying controlled-SWAP operations. It would be desirable to deduce the logarithmic negativity approximately after measuring only a few moments of the density matrix. In [37], a neural network was trained to map low-order moments (e.g. only up to the third moment) to the logarithmic negativity. In such a setting, the choice of training samples (in this case, quantum many-body states) is crucial. The authors trained on random states of two different varieties (area-law and volume-law states). The resulting network was able to perform surprisingly well on numerically simulated quantum many-body states arising in realistic dynamics.

## 4.2 Choosing the smartest measurement

Rather than merely interpreting a given set of measurement outcomes, one can strive to choose the most informative measurements possible. Given a sequence of prior observations, the observable for the next measurement can be selected so as to maximize the information.

In other words, we are looking for an adaptive-measurement strategy (or "policy"). This represents a high-dimensional optimization problem. Machine learning tools can help discovering such policies.

**Adaptive phase estimation** – The following illustrative and important pioneering example [38] was already investigated before the recent surge of interest in applications of machine learning. Consider the task of estimating an unknown phase shift inside an interferometer. For $N$ independent photons, the phase uncertainty scales as $1/\sqrt{N}$, the so-called "standard quantum limit" of phase estimation. However, by injecting an entangled state into the interferometer, one can improve on that bound, down to the Heisenberg limit $1/N$. An adaptive scheme consists in measuring one photon at a time and using all the previous results in order to select the next measurement basis. In the interferometer case, the measurement basis is imposed via an additional, controllable, deterministic phase shift. The adaptive policy is thus described by a "decision tree", in which each branch (corresponding to a sequence of

measurement outcomes) leads to a different choice of measurement basis. A search for the best policy is a search over all such trees – a formidable problem: the tree itself already contains a number of leaves that scales exponentially in the number of photons, and for each leaf a different value of the measurement basis (phase shift) can be selected. In [38], this high-dimensional optimization problem was tackled by the use of "particle swarm optimization", which is an efficient technique comparable to genetic algorithms and simulated annealing. The resulting strategies, for photon numbers up to $N = 14$, were able to beat the best previously known adaptive scheme.

Recently, an experimental implementation of these ideas was presented in [39], although restricted to $N$ independent photons (instead of an entangled multi-photon state). The authors of [39] compared particle-swarm optimization to Bayesian approaches. The latter are somewhat simpler, in that they update the probability distribution over the unknown phase shift after each new incoming measurement result according to the Bayes rule. The adaptive Bayes approach then seeks to select a measurent basis that would minimize the expected variance. In essence, this represents a "greedy" strategy, where each individual step is optimized (which need not necessarily lead to the best overall result, generally speaking).

**Experimental device characterization** – The challenge becomes more pronounced if the relationship between the observations and the underlying parameter(s) is more complex, possibly even not accessible analytically (in contrast to the situation in phase estimation, where the relation between phase shift and measurement probability is simple).

In general, we might be able to control a few parameters $V_1, V_2, \ldots$ (which correspond to the choice of measurement basis in the example above). Furthermore, the measurement result $I$ also depends on some hidden but fixed underlying model parameters $\lambda_1, \lambda_2, \ldots$, which we might want to extract as well as possible (the unknown phase shift in the example above). The mapping from the controllable parameters and the model parameters to the measurement result may be simulated numerically but is complex and not easily inverted. One important question in this setting, just as before, is "where to measure next".

This problem was studied recently experimentally for the first time using advanced machine learning techniques [40]. The setting was semiconductor quantum dots, although the principles illustrated there are fairly general. The controllable parameters are the gate and bias voltages applied to the device. The underlying model itself is specified by the actual physical device, with its detailed (unknown) potential landscape that electrons inside the quantum dot experience, together with a set of further voltages that are kept fixed during an experiment. The current $I(V_1, V_2)$ through the quantum dot depends on all of these aspects.

The authors of [40] adopt a measure of predicted "information gain" to select the "best" voltages $V_1, V_2$ for the next measurement. The information gain at any selected location in voltage space is defined as the Kullback-Leibler divergence between the probability distributions before and after the new measurement, averaged over all possible underlying "ground truths" (averaged according to the present distribution). Since we are talking about probability distributions over the space of all possible current maps $I(V_1, V_2)$, it is impossible to handle them explicitly.

In order to make progress anyway, a technique is needed to sample from the correct distribution that Bayes predicts given all previous measurements. The general technique exploited in [40] is known as "conditional variational autoencoder". A variational autoencoder is similar to the autoencoder discussed previously in these lecture notes, except that it learns to produce neuron values in the bottleneck layer that are distributed according to a fixed Gaussian distribution. The benefit of this is that feeding the encoder with random Gaussian-distributed values will generate outputs that are distributed according to the correct underlying distribution. A conditional variational autoencoder takes this one step further by allowing for the specification of extra features. In [40], training was undertaken using both simulated and

measured current maps.

## 4.3 Discovering better control sequences and designing experimental setups

In quantum control, the goal is typically to implement a desired unitary as well as possible. Standard numerical techniques exist for this purpose, with one of the most well-known being GRAPE (gradient ascent for pulse engineering; [41]). However, recently reinforcement-learning (RL) type techniques have been applied to this challenge. Even though in this context they do not yet use the full power of RL, in that they do not search for feedback-based strategies, they turn out to be a very useful addition to the control toolbox. In contrast to techniques like GRAPE, modern RL techniques are "model-free", i.e. the algorithm itself applies independently of the underlying dynamical model. In addition, adopting RL-based control makes it very easy to benefit from the most recent advances in the field of machine learning.

**State preparation** – One important task is to drive a quantum system from an initial state to a target state. In [42], the authors studied how RL (Q learning) finds protocols for this purpose, in non-dissipative systems, focussing on "bang-bang" type protocols (which are also often used to combat slow noise acting on qubits). One particular specialty of their work is the study of the "landscape" of learning: how likely is it that the RL algorithm gets stuck? They discover a spin-glass type phase transition as a function of the prescribed duration of the protocol, where it becomes hard to find an optimal protocol. Moreover, going beyond control of a single qubit, they show that the same techniques very successfully also apply to spin chains where the Hilbert space is exponentially large, yet the number of control parameters remains small.

**Control of dissipative qubits** – More advanced modern RL techniques have also been applied recently to find continuous control sequences for dissipative few-qubit systems; for first examples see [43, 44].

**Discovering experimental setups** – In a setting like quantum optics, the sequence of time-dependent control pulses is replaced by a sequence of optical devices through which photons will pass. In [45], RL has been successfully used to search for "good" setups composed of beam-splitters, prisms, holograms, and mirrors, placed on an optical table, where the input state is an entangled state generated by parametric down-conversion, and the final state is produced by measurement and post-selection. This followed an earlier pioneering work [46] tackling the same challenge with a direct automated search. In contrast to the tasks mentioned in the other examples above, there is not simply a pre-assigned target unitary to reach. Rather, a high reward is assigned to setups producing photonic states with a large degree of high-dimensional multi-partite entanglement. Remarkably, the algorithm discovers useful novel building blocks that can be inspected and analyzed afterwards. The RL technique used in [45] is called "projective simulation" [47], representing a lesser known recent alternative to the approaches discussed in these lecture notes.

## 4.4 Discovering better quantum feedback strategies

As we have explained in the previous section on reinforcement learning (Sec. 3.1), the basic paradigm involves an "agent" interacting with an "environment". This interaction goes both ways. Not only is the agent permitted to act on (control) the environment, but it can also observe the consequences and adapt its future moves accordingly. This second aspect represents *feedback*, applied directly during the interaction with the environment. It is to be distinguished from the other type of feedback which is used only during training, when the reward controls the update of the agent's strategy.

The examples of RL mentioned in the previous section still do not incorporate (direct) feedback, with their goal rather being to find an optimal control sequence that does not require

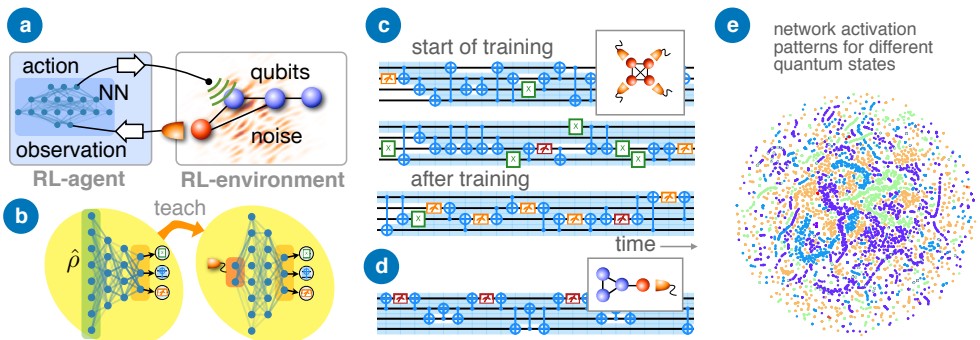

Figure 7: Discovering quantum error correction strategies from scratch [48]. (a) The setting: A neural-network-based agent controls a few qubits, applying quantum gates and measurements, with the aim of protecting the quantum information against noise. (b) RL is used for training a powerful first network that receives the quantum state $\hat{\rho}$ as input at every time step. This is then used for supervised training of a second network that only obtains the measurement results and which can be deployed in an experiment. (c,d) Quantum circuits (i.e. action sequences) for two different qubit setups. The network learns to encode the quantum information, apply periodic collective measurements (parity detection), and eventually also to correct any errors. (e) Visualization of the network activation patterns. Each point corresponds to one quantum state (reached at a particular time, in one of many trajectories). Its location is a 2D nonlinear projection (using the t-SNE method [49]) of the high-dimensional vector of neuron activations obtained in the network for that quantum state. Different clusters (colored according to the action suggested by the network) belong to quantum states that are considered qualitatively different by the network. Using t-SNE with a higher 'perplexity' parameter (here: 30) results in more clearly separated clusters, see [48].

any adaptation to the unpredictable behaviour of the environment. In other words, the optimal sequence does not contain any conditional branches.

This changes as soon as the agent is allowed to observe the quantum environment. The stochastic measurement outcome must then be considered by the agent in deciding on its subsequent actions. We will now summarize the first work applying neural-network-based RL to quantum physics including feedback [48].

Quantum feedback is an important technique for quantum technologies. If both the quantum system and the measurement are linear, many analytical results exist to help find the optimal feedback protocol. However, for nonlinear quantum systems, feedback protocols must become more complex and this is where RL can be useful. We also note that the feedback aspect represents one of the most important conceptual differences between RL and other numerical methods like GRAPE [41].

Among the possible applications of quantum feedback to nonlinear quantum systems, quantum error correction (QEC) is particularly important. The typical idea in QEC, established in the 90s by Shor and others, is to encode a "logical" qubit state into a complicated entangled multi-qubit state. This multi-qubit state then is effectively more robust to noise, in that an error can be detected and corrected. Importantly, the error detection can be performed without detecting the state of the logical qubit. While textbooks provide the useful encodings and associated error syndromes for such stabilizer codes, it is not clear for any given actual hardware what might be the most efficient way to achieve this abstract task. In addition, given a certain

hardware layout for a quantum memory device, it may turn out that some low-level, hardware-centric approaches are more efficient. For example, if the noise is spatially or temporally correlated, techniques like decoherence-free subspaces or dynamical decoupling can be very helpful.

This provides a suitable challenge for RL: Start by providing the layout of a few-qubit device, specify the qubits' connectivity and the available native quantum gates and possible measurements. Then, RL can help to find the best strategy to protect a logical qubit state from decoherence, given the noise processes acting on the device.

In our work [48], we showed how RL (natural policy gradient) can discover from scratch such quantum error correction strategies involving feedback (Fig. 7). The network finds concepts such as entangled multi-qubit states for encoding, collective qubit measurements, adaptive noise estimation, and others. None of these ideas had been provided to the network in advance.

For example, given a system of four qubits, RL automatically figures out that it is beneficial to encode a logical quantum state (first present in one of the four qubits) into a 3-qubit state, effectively re-inventing Shor's repetition code for this example. It then understands that direct measurements on any of the three code qubits are destructive, but the fourth qubit can be treated as an ancilla, such that a sequence of two CNOTs and a measurement then implements parity detection, which helps signal errors. Finally, the network develops an adaptive strategy, where after detection of an error it learns to quickly pinpoint where exactly the error occured and how to correct it.

Depending on the layout of the qubit device (e.g. which qubits can be connected via a CNOT), and depending on the available gates and the properties of the noise, the network will vary the strategies. However, the range of applicability is far wider than stabilizer codes. As RL is completely general and works without any human-provided assumptions, the same neural network can also discover completely different strategies. For example, as we show [48], in a scenario where several qubits are subject to a fluctuating field that is spatially homogeneous, the network finds a strategy where some of the qubits are observed repeatedly to gain information about the noisy field – which can then be used to correct the qubit where the quantum information is stored. The observation strategy is even adaptive, in that the network chooses a measurement basis that depends on the full sequence of previous measurement outcomes, to enhance the accuracy.

Despite the power of RL, we found that this challenge cannot be solved without any extra insights. In our case, we invented a new quantity, "recoverable quantum information", that measures the amount of quantum information that survives in a complicated entangled multi-qubit state and could, in principle, be extracted. This then serves as an immediate reward function for RL, and it is much more powerful than only calculating the overlap between the initial state and the final state after the full sequence of 200 time steps. In addition, we devised a "two-stage learning" scheme. In a first step, RL is used to train a network that is made more powerful by allowing it to see the full quantum state at any given time step. In a second step, the first RL-trained network is used to train a second network in a supervised manner, which then learns to mimic the strategy. However, this second network only receives the measurement results as input. Thus, it could be realistically deployed in an experiment, where the full quantum state is of course not available. These two key insights represent domain-specific human input. Making use of such knowledge for RL is permissible, as long as the resulting algorithm does not become restricted to special use cases but still covers a wide range of possible scenarios. Here, it covers quantum error correction for all possible settings of few-qubit quantum memories.

In the future, similar RL approaches could be applied to other physical systems with specific requirements (e.g. ion trap chips, where one may shuffle the ions between different registers;

this would represent another RL action), for finding fault-tolerant unitaries, and to treat quantum information storage in hybrid systems, where qubits are, e.g., coupled to cavities. Implementing the RL scheme experimentally will likely require dedicated hardware, like FPGAs, in order to be sufficiently fast in deciding on the next action.

# 5 Towards Quantum-Enhanced Machine Learning

Quantum algorithms promise spectacular speedups for certain tasks like factoring and search. It is therefore natural to ask whether they can also help with machine learning. We want to stress right away that even on the theoretical level there is, at the time of writing, not yet any completely compelling example of evident practical relevance for quantum-accelerated machine learning. Nevertheless, there are first insights and proposals, e.g. for quantum-accelerated linear algebra subroutines that may help with machine learning tasks [4], as well as for possible speed-ups in reinforcement learning (via Grover search), for modeling the statistics of quantum states via quantum Boltzmann machines, and for various other tasks. We can only scratch the surface of the rapidly developing literature here, and we refer the reader to a number of excellent reviews for a more complete overview [3, 4, 6].

We will start by mentioning one of the main roadblocks for a naive approach to quantum-accelerated machine learning.

## 5.1 The curse of loading classical data into a quantum machine

We think of machine learning as a way to learn from, and discover patterns in, large amounts of data. Typically, we would have in mind classical data, obtained from databases or by measurements. This immediately gives rise to a severe challenge that affects many potential quantum-accelerated algorithms if their purpose is to act on large amounts of classical data. If the quantum algorithm's complexity scales better than linear in the size $N$ of data, then this advantage will be destroyed by the need to load all the $N$ data points into the quantum machine.

That challenge can be illustrated in many examples, but let us just consider briefly the quantum Fourier transform, because we will need it later on anyway. This is a unitary operation that implements the Fourier transform on the set of $N = 2^d$ amplitudes in a $d$-qubit wavefunction. It is most well-known for its use in Shor's algorithm. To write it down, we label the basis states $|x\rangle$ by integer numbers $x = 0 \dots 2^d - 1$. These numbers can be decomposed into binary representation $x = x_0 + 2x_1 + 4x_2 + 8x_3 + \dots$, and $x_m = 0, 1$ is interpreted to be the state of qubit $m$ in the basis state $|x\rangle$. Then the quantum Fourier transform is the unitary given by

$$\frac{1}{\sqrt{N}} \sum_{k,x} e^{-ikx} |k\rangle \langle x| . \tag{53}$$

This means the coefficient of basis state $|k\rangle$ after application of the qFT is indeed the Fourier transform of the original coefficients, $\frac{1}{\sqrt{N}} \sum_x e^{-ikx} \langle x| \Psi\rangle$.

In its original implementation, the qFT needed $\mathcal{O}((\log N)^2)$ CPHASE gates, but this complexity has been improved by now to $\mathcal{O}(\log N \cdot \log \log N)$. In any case, that is exponentially faster than the classical fast Fourier transform, which needs $\mathcal{O}(N \log N)$ operations.

Unfortunately, there is no way to build a quantum sub-processor that takes, say, $2^{30} \sim 10^9$ complex numbers, stores them into a 30-qubit wave function, executes the qFT in only $\mathcal{O}(30 \log 30)$ steps, and then returns those numbers. The whole operation would be dominated by the need to load $10^9$ numbers into the sub-processor (and that is not even speaking of the

challenge of reading them out, which is impossible by measurement on a single copy of the state!).

On the other hand, Shor's algorithm does exploit the qFT to obtain a real exponential advantage over a classical computer. The trick is, of course, that the amount of data to be loaded is very modest (one single big number to be factorized), and the exponentially large set of complex amplitudes that the qFT operates on are generated internally. This points a way towards applications of quantum machine learning that are not subject to the curse of loading classical data: try to find quantum data on which to operate!

## 5.2 Quantum Neural Networks

When talking about quantum accelerated machine learning, the obvious first question is what would be the quantum generalization of an artificial neural network. Several aspects arise.

On the positive side, there is an obvious conceptual link, in that the simplest version of a neuron would be a binary neuron, which can be either in its 'resting' state (0) or in an activated state (1). This could be directly translated into a qubit, which now can also be in a superposition of such states – and multiple qubit-neurons could be in an entangled multi-qubit superposition state.

However, beyond that, there are essential problematic differences. First, the typical artificial neural network is an *irreversible* information processing device: multiple inputs may lead to one and the same output (that is obvious for image labeling, where many different images will obtain the label 'cat'). On the other hand, if one tries to exploit the power of quantum mechanics, a quantum neural network should presumably be isolated and coherent. In that case, the dynamics is reversible (unitary). Thus, we cannot immediately build a quantum version of the usual feedforward neural network. Nevertheless, one may try to create a quantum version of *reversible* classical neural networks (which are constructed such that the mapping between input and output is bijective). Alternatively, the dynamics could be made partially dissipative (e.g. by intermediate measurements or coupling to a bath) – which brings up the challenge to demonstrate that under these conditions there is still a quantum advantage.

Second, while the linear steps in a classical neural network (matrix-vector multiplication for each layer) seem to straightforwardly correlate to unitary, linear quantum dynamics, the *nonlinear* character of classical neural networks is essential. This essential nonlinearity can be built into the quantum nonlinear network via multi-qubit interactions, so that the network has sufficient power (notwithstanding the fact that the unitary dynamics in the multi-qubit Hilbert space is still linear, just as in a quantum computer). Alternatively, nonlinearity could be generated by measurements and feedback into the quantum device based on those measurements, though this disrupts the quantum coherence.

Third, a naive application of a hypothetical fully coherent quantum artificial neural network to a quantum superposition of inputs would just result in a quantum superposition of outputs. The final measurement of the output would then collapse this superposition to a single output state. Therefore, the device would simply act like a classical neural network that spits out the answer to a randomly selected input state. No speed-up would ensue. This, of course, is the general reason why it is so hard to come up with good quantum algorithms – it is not sufficient to naively rely on quantum parallelism.

## 5.3 The quantum Boltzmann machine

Earlier, we discussed the (classical) Boltzmann machine which can learn to reproduce the statistical properties of a large training data set and produce new samples according to this distribution. Can one do the same for "quantum data"?

In quantum mechanics, one way to deal with an ensemble of statistically sampled quantum states is to represent it by a density matrix. We could, therefore, ask for a "quantum Boltzmann machine" (QBM) which is able to "learn" the density matrix $\hat{\rho}$ of a given quantum state [4, 50, 51]. Presumably, for the challenge to be interesting, we are trying to learn the state of a quantum many-body system. To make this happen, we assume the QBM is in a thermal state $\hat{\sigma}$, and it obeys a Hamiltonian that has sufficiently many tuneable parameters, such that

$$\hat{\sigma} = \frac{e^{-\sum_j w_j \hat{H}_j}}{\text{tr}[e^{-\sum_j w_j \hat{H}_j}]}. \tag{54}$$

Here, the inverse temperature has been absorbed into the definition of the weights $w_j$ (meaning $w_j = \beta w_j^{\text{physical}}$). We need some measure of the deviation between target $\hat{\rho}$ and QBM state $\hat{\sigma}$, to serve as our cost function. One option [50] is the relative entropy,

$$S(\hat{\rho} \parallel \hat{\sigma}) = \text{tr}[\hat{\rho} \ln \hat{\rho}] - \text{tr}[\hat{\rho} \ln \hat{\sigma}]. \tag{55}$$

We will apply gradient descent. The derivative with respect to the weights is

$$\frac{\partial}{\partial w_j} S(\hat{\rho} \parallel \hat{\sigma}) = \text{tr}[\hat{\rho} \hat{H}_j] - \text{tr}[\hat{\sigma} \hat{H}_j]. \tag{56}$$

Thus, to update the weights, we need to measure the expectation values of the observables $\hat{H}_j$ in the target state (once), as well as in the QBM state (repeatedly, since the weights are evolving during training). If we think of the QBM as a quantum spin system, then some of the $\hat{H}_j$ might be single-spin operators (with the corresponding $w_j$ as effective magnetic fields) and others might be two-spin operators (with $w_j$ denoting a coupling constant). Longer-range interactions will allow more expressive freedom for the QBM. Provided that we do not need exponentially many tuneable parameters to achieve a good approximation, there will be the usual quantum speed-up of a quantum simulator: the QBM will yield the expectation values exponentially faster than a classical computer would be able to compute them.

Implementing a QBM in this way, to approximate an interesting quantum many-body state, is still a formidable challenge. For example, if we implement longer-range couplings in order to make the QBM more powerful, then we also need to be able to measure the corresponding two-point correlators in the target state. In addition, it may not even be clear in the beginning how to most effectively establish a correspondence between the degrees of freedom in the target system Hilbert space and the degrees of freedom of the QBM. Such a correspondence has been assumed implicitly in writing down the expression for the cost function above, since $\hat{H}_j$ must be able to act on both Hilbert spaces. In practice, we will have to set up a 'translation table' that determines, e.g., which spin operator in the QBM relates to which operator in the target system.

## 5.4 The quantum principal component analysis

One example of quantum data that is typically hard to analyze is a quantum many-body state, expressed via its density matrix $\hat{\rho}$, which is exponentially large in the number of degrees of freedom. Is there a way to analyze it with the help of quantum subroutines? For example, can we decompose it into its eigenvectors and study the most important ones, i.e. those with the largest eigenvalues?

The answer is yes, and the tool invented for this task is called the quantum principal component analysis (qPCA) [52]. It is a nice example that illustrates the power of quantum-accelerated data processing – and also the range of tricks from the quantum computation

toolbox that go into the construction of such an algorithm. We will now indicate the main steps.

One way of obtaining the eigenvalues and -vectors of a (Hermitean) matrix $\hat{\rho}$ on a classical computer would be to consider the exponential $e^{-i\hat{\rho}t}$ and Fourier-transform it with respect to time. Since $\hat{\rho} = \sum_l p_l |v_l\rangle \langle v_l|$ in its eigenbasis, we have $e^{-i\hat{\rho}t} = \sum_l e^{-ip_l t} |v_l\rangle \langle v_l|$, and the Fourier transform would be $\frac{1}{2\pi} \int_{-\infty}^{+\infty} dt\, e^{i\omega t} e^{-i\hat{\rho}t} = \sum_l \delta(\omega - p_l) |v_l\rangle \langle v_l|$. The eigenvalues can then be read off from the resonance peaks in this Fourier transform, and their "weight" is the projector onto the eigenvector. Even a Fourier transform over a finite time-range $t$ will be able to resolve eigenvalues that are further apart than $1/t$. Of course, this algorithm is nowhere near as efficient as the best classical algorithms for matrix diagonalization of a $N \times N$ matrix, but it is a feasible method.

The basic idea of qPCA is to take this method and accelerate it via the quantum Fourier transform. However, this first requires us to produce $e^{-i\hat{\rho}t}$, the exponential of a density matrix, on a quantum machine!

One elementary but important observation is that the eigenvalues (and -vectors) of any matrix $\hat{\rho}$ are nonlinear functions of the elements of that matrix. In the present context this means there is no way to apply a (fixed) unitary to an arbitrary $\hat{\rho}$ and end up with its eigenvalues and -vectors, since that would be a linear operation. Indeed, the exponential $e^{-i\hat{\rho}t}$ mentioned above is nonlinear in $\hat{\rho}$. This means our quantum machine will have to operate on states that are already themselves nonlinear in $\hat{\rho}$, i.e. of the form $\hat{\rho} \otimes \hat{\rho} \otimes \hat{\rho} \otimes \ldots \otimes \hat{\rho}$ – we will therefore necessarily need multiple identically prepared copies of the state. If $\hat{\rho}$ is the state of a quantum many-body system, multiple copies of this system (with identical parameters) will have to be prepared, which requires some experimental effort.

Consider the unitary $e^{-i\hat{\rho}t}$ acting on some state $\hat{\sigma}$, i.e. try to compute $e^{-i\hat{\rho}t} \hat{\sigma} e^{+i\hat{\rho}t} \approx \hat{\sigma} - it[\hat{\rho}, \hat{\sigma}] + \ldots$. The crucial trick introduced in Ref. [52] is the realization that this can be obtained to leading order by performing an exponential SWAP operation on a product state of $\hat{\rho}$ and $\hat{\sigma}$:

$$\mathrm{tr}_1 e^{-i\hat{S}\Delta t} \hat{\rho} \otimes \hat{\sigma} e^{+i\hat{S}\Delta t} = \hat{\sigma} - i\Delta t [\hat{\rho}, \hat{\sigma}] + \mathcal{O}(\Delta t^2). \tag{57}$$

Here $\hat{S}$ is the SWAP which operates on two subspaces by $\hat{S} |i\rangle \otimes |j\rangle = |j\rangle \otimes |i\rangle$, and $\mathrm{tr}_1$ is the partial trace over the first subsystem (i.e. we discard this system after the operation and will never measure it). The density matrix $\hat{\rho}$ describes the quantum many-body system and $\hat{\sigma}$ is the state of the quantum computer. We need to be able to do partial swaps $e^{-i\hat{S}\Delta t}$ on corresponding pairs of qubits of both these systems. Again, this is not trivial experimentally, since it presumes, e.g., that these operations can be carried out fast enough that the many-body system does not evolve (or its dynamics has to be frozen, e.g. by setting couplings to zero). Repeated application of the trick in Eq. (57) to a state $\hat{\sigma} \otimes \hat{\rho} \otimes \hat{\rho} \otimes \hat{\rho} \otimes \ldots$ will result in a higher-order approximation to $e^{-i\hat{\rho}t}$, where we have to apply the exponential SWAP (and the partial trace) separately to each of the multiple copies of $\hat{\rho}$.

We now want to exploit the qFT. To this end, we do not just need $e^{-i\hat{\rho}t}$ for one particular value of the time $t$, but for a whole time interval. Moreover, for the qFT, the whole time trace has to be in the quantum memory simultaneously (as opposed to repeatedly running the quantum computer for different values of time $t$). The way this is done is to set up some auxiliary degrees of freedom that are in a superposition of states $|n\Delta t\rangle$ which label time (the $n$ would be an integer, and the encoding would be done in the way we discussed for the qFT above). Afterwards, one would apply the exponential of $\hat{\rho}$ in the manner discussed above, but *conditioned* on the auxiliary state. This results in:

$$\sum_n |n\Delta t\rangle \otimes e^{-i\hat{\rho}n\Delta t} |\chi\rangle, \tag{58}$$

where $|\chi\rangle$ was the original state of the quantum computer. This is an entangled state, entangling the "time-label states" with the corresponding time-evolved states. To do this, one has to perform conditional SWAP gates.

Finally, one can apply the qFT to this state. Note that instead of simple complex amplitudes we now have a quantum state attached to each time bin. The qFT will result in the spectrum, which is peaked near the eigenfrequencies, with the corresponding eigenvectors attached. Considering the special case where the initial state is $\hat{\rho}$ itself, one obtains [52] for the final state of the quantum computer (now again writing everything as a mixed state):

$$\sum_j p_j \left|\tilde{p}_j\right\rangle\left\langle\tilde{p}_j\right| \otimes \left|v_j\right\rangle\left\langle v_j\right|. \tag{59}$$

Here the $p_j$ are the eigenvalues of $\hat{\rho}$, $\left|v_j\right\rangle$ are its eigenvectors, and $\left|\tilde{p}_j\right\rangle$ represent vectors peaked around the eigenvalues $p_j$ in the Hilbert space that represents the frequencies after application of the qFT.

One can now sample by measurements from this state, to obtain properties of the eigenvalues and -vectors of the density matrix. A measurement of the first Hilbert space (where the $\left|\tilde{p}_j\right\rangle\left\langle\tilde{p}_j\right|$ live) will project the overall state down to a random eigenvector, but with larger probability for the more important ones (where $p_j$ is larger). Afterwards, arbitrary properties of the eigenvector $\left|v_j\right\rangle$ can be measured.

As shown in [52], the overall running time of qPCA grows polynomially in the number of particles, rather than the exponential growth of effort that "brute-force" normal quantum state tomography would require. There are of course multiple challenges for the qPCA: one needs multiple copies of the many-body quantum state, to produce, via controlled SWAP operations, a single copy of one (randomly selected) eigenstate, on which one can then perform a few measurements (of commuting observables). Afterwards, the whole procedure has to be repeated again on fresh copies, to find out more on some other randomly selected eigenstate. And if the original data is presented in classical form (rather than a quantum many-body state), one runs into the bottleneck mentioned above. In this context, it is important to mention that qPCA was one of the first quantum algorithms for which a quantum-inspired classical counterpart was found recently [53]. The scenario assumed there is to have "sampling" access to classical data, i.e. be able to obtain a particular component of a vector, at random with a certain probability prescribed by the vector. This then leads to a stochastic algorithm that does not require exponential effort.

## 5.5 Quantum reinforcement learning

If both the agent and the environment are quantum, one can imagine a fully quantum-mechanical version of reinforcement learning. In the simplest, direct translation from the classical domain, we would have an agent+environment quantum device that proceeds through all training trajectories simultaneously (i.e. all sequences of training epochs, with all possible evolutions of the reward). However, once we measure the agent, we would be left in only one branch, and overall there would be no gain in time needed to reach this reward level.

The question is, therefore, how to exploit some quantum algorithm for RL. The most famous quantum algorithm, Shor's factoring, with its exponential acceleration due to the quantum Fourier transform, is relatively specialized. By contrast, Grover's algorithm for search among $N$ items, has been a very useful starting point for diverse applications. Luckily, RL in its simplest incarnations can be viewed as a search problem that may benefit from Grover's scheme, which accelerates search from the classically expected $\mathcal{O}(N)$ steps to $\mathcal{O}(\sqrt{N})$ steps. While this may not seem much, compared to exponential acceleration, it can still be substantial

if the number $N$ of database entries is large (e.g. imagine $N \sim 10^{12}$, leading to a millionfold acceleration!).

Imagine a simplified toy RL-problem, where only precisely one "good" sequence of actions yields a reward of 1, while all other sequences yield reward 0. This is directly a search problem, where the action sequences can be taken as the entries in the database, and the reward is the function used to label the "good" entry in Grover's algorithm. If we have a quantum environment that can yield the reward given an arbitrary action sequence, then it can be used as the quantum oracle in Grover's algorithm, accelerating the RL search. This is the basic idea exploited in [54].

# 6 Conclusions

Applications of machine learning, especially deep learning, to physics are now appearing rapidly, in many different topics. Quantum devices provide a particularly fertile area of applications, from the analysis of measurement data to the optimization of control strategies, and eventually such devices in turn might help to accelerate machine learning itself. Crucially, any of these applications automatically benefits from the astonishing speed of progress in the machine learning community: building blocks like reinforcement learning strategies can be easily substituted by more powerful variants, and many of the cutting-edge machine learning algorithms quickly become available in the form of relatively easy-to-use code. It is an exciting time to be exploring just how far we can take this approach to doing physics! I do hope that the basics provided in these compact lecture notes will provide you with a good starting point to enter the field.

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
