# Peer review of "Machine Learning and Quantum Devices"

_SciPost Physics Lecture Notes, doi:SciPost Phys. Lect. Notes 29 (2021)_

## Round 1 · Referee Report · Lukas Grünhaupt (Referee 1) · 2021-2-20

Strengths

  • Easily accessible general level introduction to machine learning geared towards physicists
  • Relating /illustrating the concepts of neural networks to concepts in physics, i.e. cost function <-> Hamiltonian, ...
  • Describing algorithms not only mathematically but also giving an intuitive picture.
  • Clearly explained simple examples for different machine learning approaches
  • Provides a starting point for literature about machine learning applications for quantum information devices, and experiments with such devices harnessing machine learning techniques
  • Outlook and discussion of possible use of quantum computing algorithms and devices for machine learning for more advanced readers

Weaknesses

  • Potentially too basic of an introduction for people with some prior knowledge in the field
  • The introductory chapter could benefit from additional references for further reading, where more details of the presented algorithms are discussed

Report

The lecture notes Machine Learning and Quantum Devices encompass an easily accessible introduction to the field and nomenclature of machine learning, a sample overview of experiments and theoretical proposals to harness the capabilities of machine learning for quantum information devices to improve readout, control and error detection/mitigation based on physical device parameters, and an outlook on potential applications where machine learning algorithms could benefit from quantum computing algorithms. As a reviewer without any expertise in the field of machine learning I find the introductory chapter is well structured and provides an easily accessible introduction to the basics of machine learning. I especially liked the examples clearly illustrating the algorithmic approaches. The links and reference to a more complete set of lecture notes by the same author as well as the link to a code repository with examples to get working code is highly appreciated.
The second part of the notes serves as an illustration and inspiration where and how machine learning can be applied to quantum information devices. In the final chapter the author provides a balanced discussion of the promises, but also pitfalls of quantum computing for the field of machine learning.
Given the rather small number of lectures (4) dedicated to the topic, the author can of course not dive too deep into the different topics. However, thanks to the clear introductory chapter and some links to literature the lecture notes provide a good starting point and definitely an inspiration for the reader to further look into the topic of machine learning for quantum devices.

Requested changes

  • First paragraph on page 5, "Interestingly, practically any nonlinear activation function will do the job, although some may be better for training than others. However, a representation by multiple hidden layers may be more efficient (defining a so-called “deep network”).": Could the author further expand on what more efficient would mean in this context.
  • Figure 2: I think it would improve readability to move panels d)-h) into a separate figure and place this figure closer to the discussion of CNN, maybe page 14/15
  • Page 21, 3rd paragraph: "The resulting training progress is shown in Fig. 4c." the reference appears to be misplaced as Fig. 4c shows a walker reaching a target, but that is only discussed later than the reference in the text.
  • Figure 5 caption: d) needs to be changed to c)
  • Page 32 4th paragraph: [...] In each step, has as input available the information from the encoder as well as the sequence of words [...]. It appears an 'it' is missing?
  • Reference [28] seems not formatted correctly
  • Ref. [33] is also published in PRX: https://doi.org/10.1103/PhysRevX.10.011006

---

## Round 2 · Author Response

Dear editor, dear referee,

I very much appreciate the feedback, and I apologize for the delay.

Best regards,
Florian Marquardt

---

## Round 2 · List of Changes

Replies to referee:

Thank you very much for your effort and good suggestions, and I apologize for the delay in implementing the revisions.

  • I now made it more explicit what 'more efficient' means for a deep network: “However, a representation by multiple hidden layers may be more efficient, i.e. would be able to reach a better approximation with the given overall number of neurons or use fewer neurons for a given approximation accuracy (sometimes this difference can be dramatic). Such a multi-layer network is sometimes called a “deep network”, especially if the number of layers becomes larger than a handful.”

  • Thank you for the suggestion regarding figure 2. However, panel (d) also belongs to CNN, and I did not find a good way to separate off the (small) autoencoder part or the small image recognition part (a) [given the format of figures in scipost]. So I left it as is.

  • "The resulting training progress is shown in Fig. 4c." -> I have now corrected this, it was Fig 4d.

  • Fig. 5 caption corrected.

  • page 32, “it has as input available” -> revised, inserted 'it'

  • Ref 28 fixed.

  • Ref 33 fixed.

Thank you again.

---

## Editorial Decision

published